# ED-BFN: Electron Density Point Clouds Enable High-Fidelity 3D Molecular Generation via GeoBFN

## Abstract

Designing molecules that complement the 3D shape and chemical environment of the binding pocket on their biological target is a central challenge in drug design. While current generative models often treat molecular structures as rigid—using fixed atomic coordinates or abstract ligand features—they struggle to capture the continuous, interaction-driven nature of molecular recognition. To bridge this gap, we propose leveraging *electron density (ED)* — a continuous, physics-grounded representation encoding conformational ensembles and local chemical environments. We introduce **ED-BFN**, an SE(3)-equivariant generative model that generates 3D molecular structures conditioned on sparse, pharmacophore-annotated point clouds derived from ligand ED. This approach provides a structure-aware prior without precise atom coordinates of protein structures. Unlike existing ED-based models, ED-BFN maintains strict spatial fidelity by aligning generated atoms with underlying electronic features. Evaluated on the DUD-E benchmark, ED-BFN recovers **37/101** bioactive molecules under oracle setting (with ground-truth atom count provided) and **28/101** in fully end-to-end setting (atom count predicted from electron density integral). Furthermore, **45.7%** of generated poses achieve lower docking scores compared to reference redocking poses.To our knowledge, ED-BFN is the first generative model to represent electron density condition as a point cloud, and currently achieves state-of-the-art performance on bioactive molecule recovery and docking score improvement.

## 1 Introduction

Drug design is a high-risk, multi-year, multi-billion-dollar endeavor, where computational acceleration has become a critical lever to improve success rates and reduce attrition (Qureshi et al., 2023). Structure-based drug design (SBDD) leverages 3D structural information of biological targets to guide ligand design with high binding affinity and specificity (Pant et al., 2022). However, conventional SBDD methods operate on static atomic coordinates, which is a rigid approximation that neglects biomolecular flexibility and dynamic electron distributions, leading to suboptimal designs and poor generalization in real biological environments (Renaud et al., 2018; Davis & Edge, 2017). Recent deep generative models for SBDD, such as ResGen (Zhang et al., 2023), Pocket2Mol (Peng et al., 2022), and TargetDiff (Guan et al., 2023), have shown promise but inherit the rigidity of coordinate-based inputs, suffering from poor generalization to flexible sites.

We propose a novel redefinition of the conditioning signal, transitioning from atomic coordinates to experimentally derived electron density maps. X-ray crystallography intrinsically yields electron density maps. Cryo-electron microscopy (cryo-EM) generates Coulomb potential maps, which are demonstrably comparable to electron density maps. These maps collectively provide a continuous, volumetric representation that precisely encodes time-averaged electron distributions within a molecular system (Zuo, 2004) . This comprehensive representation inherently captures conformational ensembles and local chemical environments, thereby offering a "softer" prior. Such a prior effectively bridges the flexibility of ligand-based reasoning with the spatial precision of structure-based design (Ding et al., 2022).

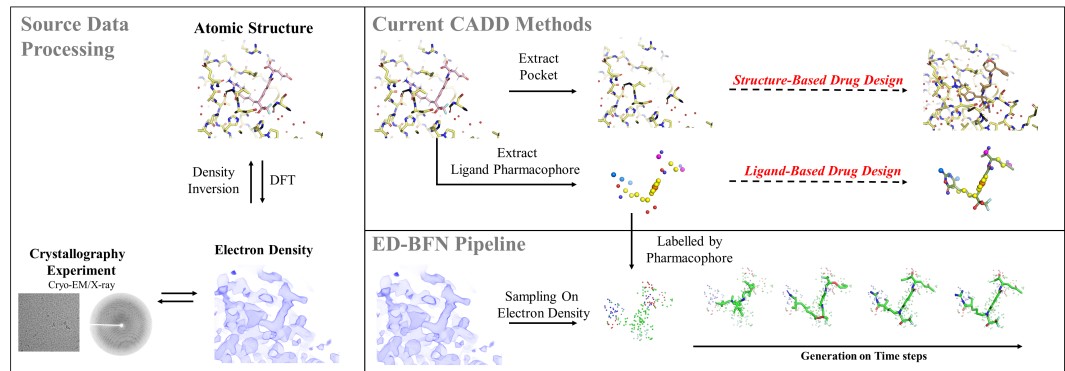

Figure 1: **From Atomic Coordinates to Electron Density Point Clouds in CADD.** *(Left)* Source data hierarchy: Experimental diffraction data yields electron density maps, from which atomic structures are derived. *(Upper)* Conventional CADD paradigms: Structure-Based Drug Design (SBDD) operates on pocket atomic coordinates; Ligand-Based Drug Design (LBDD) relies on 3D pharmacophore features. *(Bottom)* **ED-BFN**: We sample points from the electron density surrounding the ligand, label them with local pharmacophore features, and generate molecules via a GeoBFN model conditioned directly on this point cloud.

Following the emergence of the first electron-density (ED)-based molecular generation model (Wang et al., 2022), including ED2Mol (Li et al., 2025a), ECloudGen (Zhang et al., 2024), and Electron-Density2 (Parrilla-Gutiérrez et al., 2024), a shared architectural pattern has taken hold: all rely on voxelized representations of electron density as input, yet decode into molecules through pathways that compromise structural fidelity or physical realism.

Specifically, Wang's model uses GAN to generate a pocket filler ED map using the ED of protein pocket as input. This filler then conditions a Vector Quantized Variational Autoencoder (VQVAE) to sample ligand ED maps, which are subsequently filled with fragments from a library to produce final ligand structures. ECloudGen and ElectronDensity2 circumvent 3D conformational modeling entirely by regressing electron density maps to 1D SMILES strings, effectively discarding the spatial information that motivates ED-based design in the first place. ED2Mol takes a more geometrically grounded approach: it generates molecules fragment-by-fragment within the density volume, guided by a physics-inspired scoring function called Qscore (Terwilliger et al., 2006) that evaluates predicted electron density intensity and atomic partial charges. To enforce steric feasibility, it imposes strict constraints prohibiting ligand atoms from overlapping with the van der Waals radii of pocket residues. While this enables the generation of self-consistent, low-strain molecules, ED2Mol fundamentally lacks an explicit mechanism to model or optimize ligand-pocket interactions, such as hydrogen bonding, hydrophobic packing, or electrostatic complementarity. Its Qscore operates at the atomic density level, not at the interaction level. Consequently, generated molecules may "fit" the density map geometrically, but fail to engage the pocket in a biologically meaningful way.

As illustrated in Figure 1, our approach addresses these limitations by fundamentally redefining the data representation paradigm for molecular generation using ED. First, we leverage a point cloud representation for ED, moving beyond conventional voxelized inputs. Second, to enhance the potential for forming ligand-pocket interactions, we enrich the ED representation by labeling local chemical features on the surface of the point cloud. Specifically, we augment points sampled from the region where density is above the threshold of the ED with both geometric and chemical descriptors. This sparse, surface-aware representation not only preserves essential spatial structure while drastically reducing computational overhead but, crucially, enables the direct modeling of ligand-pocket interfaces. Third, eschewing existing fragment libraries, we build molecules atom-by-atom in 3D space. To realize this novel paradigm, we introduce ED-BFN, a generative model based on GeoBFN architecture and specifically designed for 3D molecule design.

Our contributions are summarized as follows:

1. We formalize the limitations of existing ED-based design methods and propose a unified view of drug design grounded in electron density as a flexible, interaction-aware prior.

2. We introduce a novel point-cloud representation of electron density, which sampled from high-density region and enriched with chemical features, enables efficient, high-fidelity conditioning without voxelization.

3. We develop ED-BFN, a molecular generation model based on the GeoBFN architecture, that generates molecules in 3D space conditioned on the point-cloud ED representation. The model explicitly encourages ligand-pocket complementarity through geometric and electrostatic alignment.

4. We demonstrate state-of-the-art performance on benchmark: ED-BFN achieves significantly higher recovery rates of bioactive conformations and better docking score improvement, which indicating generated molecules more closely resemble real drug-like binders in both geometry and interaction patterns.

## 2 RELATED WORK

**Molecular Generation Guided by Electron Density.** ECloudGen encodes the binding pocket as a 3D voxel grid and uses a variational autoencoder (VAE) to generate ligand electron density. This density is then regressed into SMILES strings, which discards all 3D conformational information. ED2Mol also operates on voxelized pockets but constructs molecules geometrically by placing molecular fragments within the predicted electron density, guided by a neural scoring function and a fragment library. Although it preserves 3D awareness, ED2Mol is limited by fragment diversity: large libraries lead to computationally expensive searches, while small libraries restrict the explorable chemical space. In addition, it does not explicitly model ligand-pocket interactions. Both approaches rely on dense voxel grids and hybrid discrete-continuous representations. We overcome these limitations by adopting a point-cloud representation.

**Geometric Deep Learning for Molecular Generation.** Due to the inherent 3D nature of molecular interactions, incorporating geometric priors into generative models has become essential. Recent advances focus on SE(3)-equivariant architectures, continuous relaxations of discrete molecular graphs, and topology-aware generation, all with the aim of producing geometrically valid and physically plausible molecules. Notable examples include **GeoBFN** (Song et al., 2024), which employs a Bayesian flow network to enable arbitrary-step sampling and generate molecules with improved geometric consistency, and **GeoRCG** (Li et al., 2025b), which leverages representation learning to produce high-quality 3D conformations. Building upon this foundation, we adapt the GeoBFN framework to operate not on atomic graphs but on point clouds derived from electron density.

## 3 PRELIMINARIES

This section introduces GeoBFN, the foundational generative framework upon which ED-BFN is built. We highlight its key distinctions from standard diffusion models, particularly its operation in a continuous parameter space.

### 3.1 PROBLEM DEFINITION: CONDITIONAL 3D MOLECULAR GENERATION

Drug design is formulated as a conditional generation task. The goal is to generate a ligand molecule $\mathcal{L} = \{(\mathbf{x}_i, z_i)\}_{i=1}^N$, where $\mathbf{x}_i \in \mathbb{R}^3$ is the 3D position and $z_i$ is the atomic type of the $i$-th atom, guided by given conditions. In conventional SBDD, this information is the pocket's atomic coordinates, such as MolCRAFT (Qu et al., 2024). In our work, the conditioning signal is a pharmacophore-labeled electron density point cloud $\mathcal{P} = \{(\mathbf{p}_j, l_j)\}_{j=1}^M$, where $\mathbf{p}_j \in \mathbb{R}^3$ is the coordinate and $\mathbf{l}_j \in [0, \mathcal{L}_{\text{pharm}})$ is the label of the j-th point.

### 3.2 MOLECULAR GENERATION IN PARAMETER SPACE WITH GEOBFN

GeoBFN adapts the BFN (Graves et al., 2023) framework for 3D molecular generation. Its core innovation lies in treating the molecule not as a fixed set of coordinates, but as a set of *distribution parameters* $\boldsymbol{\theta} = \{\boldsymbol{\theta}_i\}_{i=1}^N$. Each $\boldsymbol{\theta}_i$ parameterizes the marginal distribution of an atom — typically, a Gaussian for its position ($\boldsymbol{\theta}_i^{\text{pos}} = \{\boldsymbol{\mu}_i, \lambda_i\}$) and a categorical distribution for its type ($\boldsymbol{\theta}_i^{\text{type}}$).

The generative process involves a "sender-receiver" communication game over $T$ timesteps:

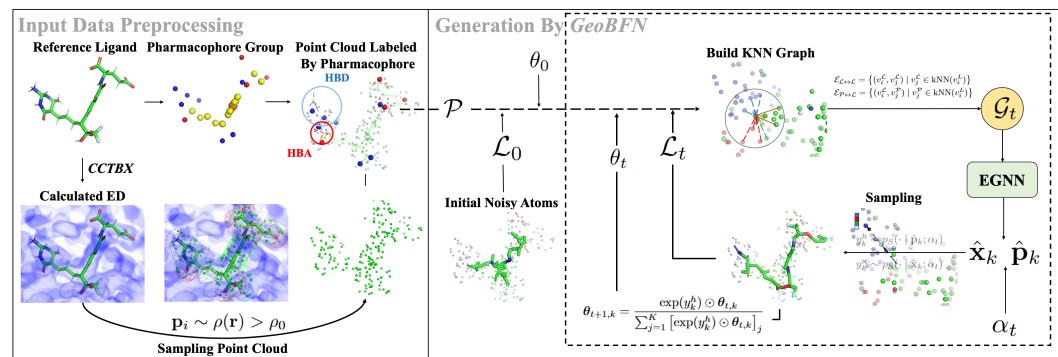

Figure 2: **Overview of the Electron Density-Guided Molecular Generation Pipeline.** Starting from the 3D electron density field $\rho(\mathbf{r})$, we construct a pharmacophore-labeled point cloud $\mathcal{P}$, which serves as a geometric and functional prior to condition the GeoBFN generative model. The generation process is guided via a dynamically constructed heterogeneous interaction graph that connects point cloud context nodes to evolving ligand atoms, enabling spatially-aware and chemically consistent molecule synthesis.

1. **Sender (Forward Process):** At step $t$, the sender takes the true molecule $\mathcal{L}$ and injects noise according to a distribution $p_S(\mathbf{y}_t \mid \mathcal{L}; \alpha_t)$, producing a noisy latent $\mathbf{y}_t$.
2. **Receiver (Reverse Process):** The receiver, parameterized by a neural network $\Phi$, takes the current belief $\boldsymbol{\theta}_{t-1}$, the conditioning signal $\mathcal{P}$, and time $t$, to predict an updated set of parameters $\hat{\boldsymbol{\theta}}_t = \Phi(\boldsymbol{\theta}_{t-1}, \mathcal{P}, t)$. This prediction defines an *output distribution* $p_O(\hat{\mathcal{L}} \mid \boldsymbol{\theta}_{t-1}, \mathcal{P}; t)$ from which a clean molecule $\hat{\mathcal{L}}$ can be sampled. The receiver then simulates what noisy observation it *would have received* if $\hat{\mathcal{L}}$ were the true molecule, yielding the *receiver distribution*:

$$p_R(\mathbf{y}_t \mid \boldsymbol{\theta}_{t-1}, \mathcal{P}; t) = \mathbb{E}_{\hat{\mathcal{L}} \sim p_O}\left[ p_S(\mathbf{y}_t \mid \hat{\mathcal{L}}; \alpha_t) \right].$$

3. **Bayesian Update:** Crucially, BFN performs a closed-form, *structured Bayesian update* to refine its belief from $\boldsymbol{\theta}_{t-1}$ to $\boldsymbol{\theta}_t$ based on the received noisy latent $\mathbf{y}_t$ and the predicted distribution. This update is deterministic and given by a function $h$: $\boldsymbol{\theta}_t = h(\boldsymbol{\theta}_{t-1}, \mathbf{y}_t, \alpha_t)$.

## 3.3 TRAINING OBJECTIVE

The model is trained to minimize the Kullback-Leibler (KL) divergence between the sender's noise distribution and the receiver's predicted distribution at each timestep. Formally, for a single molecule $\mathcal{L}$ and conditioning signal $\mathcal{P}$, the loss over $T$ timesteps is:

$$\mathbb{L}(\mathcal{L}, \mathcal{P}) = \mathbb{E}_{t \sim \mathcal{U}(1,T)}\left[ \mathbb{E}_{\mathbf{y}_t \sim p_S(\cdot \mid \mathcal{L}; \alpha_t)} \left[ D_{\mathrm{KL}}\left( p_S(\mathbf{y}_t \mid \mathcal{L}; \alpha_t) \parallel p_R(\mathbf{y}_t \mid \boldsymbol{\theta}_{t-1}, \mathcal{P}; t) \right) \right] \right],$$

This objective encourages the receiver to accurately reconstruct the clean molecule from its noisy observation, conditioned on the electron density prior $\mathcal{P}$.

## 4 METHOD: ELECTRON-DENSITY GUIDED MOLECULAR GENERATION

As shown in Figure 2, our framework leverages the 3D electron density field $\rho(\mathbf{r})$ calculated from a reference molecule, as a geometric and chemical prior to guide the generative process of novel ligands. The methodology comprises two core components: (1) construction of a pharmacophore-labeled point cloud $\mathcal{P}$ from $\rho(\mathbf{r})$, and (2) conditioning of the GeoBFN generation process via a dynamically updated heterogeneous graph that integrates $\mathcal{P}$ with the evolving ligand representation.

### 4.1 CONSTRUCTING THE PHARMACOPHORE-LABELED ELECTRON DENSITY POINT CLOUD

Starting from a reference molecule with known atomic coordinates and element types, we first compute its electron density field using cctbx (Grosse-Kunstleve et al., 2002). From this field, we

construct a 3D point cloud where each point is annotated with a pharmacophore label indicating its potential role in hydrogen bonding interactions.

The process has two intuitive steps:

1. **Sampling from High-Density Regions.** We sample $M$ points from regions where the electron density exceeds a threshold $\rho_0$. This focuses sampling near atomic nuclei — the chemically relevant zones. Formally, we sample:

$$\mathbf{p}_i \sim \rho(\mathbf{r}) \cdot \mathbb{I}[\rho(\mathbf{r}) > \rho_0],$$

where $\mathbb{I}[\cdot]$ is the indicator function.

2. **Assigning Pharmacophore Labels.** Each sampled point is labeled based on its proximity to hydrogen bond donor (HBD) or acceptor (HBA) atoms, identified using standard chemical rules. The label $l_i \in \{0, 1, 2, 3\}$ encodes:
   - **0**: No proximity to HBD or HBA atoms.
   - **1**: Close to at least one HBD atom, but no HBA.
   - **2**: Close to at least one HBA atom, but no HBD.
   - **3**: Close to both HBD and HBA atoms.

   "Close" means within the atom's Van der Waals radius.

The final point cloud $\mathcal{P} = \{(\mathbf{p}_i, l_i)\}_{i=1}^{M}$ thus captures both the shape of the electron density and the spatial layout of key interaction sites, providing a physically grounded, functionally annotated 3D representation ideal for guiding molecular generation.

### 4.2 Heterogeneous Graph Construction for GeoBFN Conditioning

Let the evolving ligand at timestep $t$ be represented as $\mathcal{L}_t = \{(\mathbf{x}_k, \mathbf{h}_k)\}_{k=1}^{N}$, where $\mathbf{x}_k \in \mathbb{R}^3$ is the 3D coordinate and $\mathbf{h}_k \in \mathbb{R}^{d_h}$ is the feature vector (e.g., atom type embedding) of the $k$-th atom. GeoBFN maintains distribution parameters $\boldsymbol{\theta}_t^{\mathcal{L}} = \{\boldsymbol{\theta}_{t,k}\}_{k=1}^{N}$, where each $\boldsymbol{\theta}_{t,k}$ parameterizes the atom's marginal distribution (e.g., Gaussian for position, categorical for type).

To condition generation on $\mathcal{P}$, we construct a heterogeneous interaction graph $\mathcal{G}_t = (\mathcal{V}, \mathcal{E})$ that dynamically links point cloud context nodes to ligand atom nodes.

**Node Set.** The node set $\mathcal{V} = \mathcal{V}_{\mathcal{P}} \cup \mathcal{V}_{\mathcal{L}}$ consists of:

1. **Point cloud nodes** $\mathcal{V}_{\mathcal{P}} = \{v_i^{\mathcal{P}}\}_{i=1}^{M}$: Each node $v_i^{\mathcal{P}}$ is initialized with a learnable embedding of its pharmacophore label: $\mathbf{f}_i^{\mathcal{P}} = \mathrm{MLP}_{\mathrm{emb}}(\mathbf{l}_i) \in \mathbb{R}^d$.

2. **Ligand atom nodes** $\mathcal{V}_{\mathcal{L}} = \{v_k^{\mathcal{L}}\}_{k=1}^{N}$: Each node $v_k^{\mathcal{L}}$ is initialized with features derived from $\boldsymbol{\theta}_{t,k}$, e.g., position mean $\boldsymbol{\mu}_{t,k} \in \mathbb{R}^3$ and log-precision $\log \lambda_{t,k} \in \mathbb{R}$ for coordinate distributions.

**Edge Set.** The edge set $\mathcal{E} = \mathcal{E}_{\mathcal{L} \leftrightarrow \mathcal{L}} \cup \mathcal{E}_{\mathcal{P} \to \mathcal{L}}$ is constructed by performing $k$-nearest neighbor search for each ligand atom node over the *combined* set of ligand and point cloud nodes, based on Euclidean distance between current position estimates $\hat{\mathbf{x}}_k = \boldsymbol{\mu}_{t,k}$ and point coordinates $\mathbf{p}_i$:

1. **Ligand–Ligand Edges** ($\mathcal{E}_{\mathcal{L} \leftrightarrow \mathcal{L}}$): Connect each ligand atom to its nearest neighbors *within the ligand set*:
$$\mathcal{E}_{\mathcal{L} \leftrightarrow \mathcal{L}} = \left\{ (v_i^{\mathcal{L}}, v_j^{\mathcal{L}}) \mid v_j^{\mathcal{L}} \in \mathrm{kNN}(v_i^{\mathcal{L}}; \mathcal{V}_{\mathcal{L}}) \right\}.$$

2. **Point Cloud to Ligand Edges** ($\mathcal{E}_{\mathcal{P} \to \mathcal{L}}$): Connect each ligand atom to its nearest neighbors *from the point cloud*, with edges directed from point cloud to ligand:
$$\mathcal{E}_{\mathcal{P} \to \mathcal{L}} = \left\{ (v_j^{\mathcal{P}}, v_i^{\mathcal{L}}) \mid v_j^{\mathcal{P}} \in \mathrm{kNN}(v_i^{\mathcal{L}}; \mathcal{V}_{\mathcal{P}}) \right\}.$$

The graph $\mathcal{G}_t$ is processed by an **equivariant graph neural network (EGNN)** backbone within GeoBFN. The EGNN maps the current ligand parameters $\boldsymbol{\theta}_t^{\mathcal{L}}$ and context $\mathcal{P}$ to updated parameter predictions $\hat{\boldsymbol{\theta}}_t^{\mathcal{L}}$, which guide the generation on next timestep. More details of the full generation loop can be viewed in Appendix Algorithm 3.

# 5 EXPERIMENTS

## 5.1 EXPERIMENTAL SETUP

**Datasets.** We trained and validated our model on a curated dataset of approximately 2.4 million drug-like molecules with defined stereochemical conformations, for which we computed the electron density. For evaluation, we used the Directory of Useful Decoys: Enhanced (DUD-E) dataset (Mysinger et al., 2012), comprising 102 diverse protein targets (kinases, proteases, GPCRs, ion channels). Each DUD-E target is associated with a PDB ID, representing one 3D protein-ligand complex, and over 200 known bioactive ligands (often without 3D structures). This makes DUD-E a standard benchmark, providing the experimentally determined 3D ligand-binding pocket structures for input and other active compounds for positive labels. Instead of explicit pocket structures, our model inputs the electron density derived from the co-crystallized ligand within the PDB ID. This input strategy is compatible with our training and implicitly conveys pocket constraints. We generated 1,000 ligands per target, assessing their ECFP4 Tanimoto similarity to known active ligands to evaluate reproduction of active compounds. Additionally, 100 generated ligands per target were randomly selected and underwent 3D conformation and binding mode evaluation.

To prevent data leakage and ensure a fair evaluation, we performed optimal alignment between the point clouds of co-crystallized ligands from DUD-E test targets and those in the training set. Only test samples with an RMSD exceeding 2.0 Å after alignment were retained, guaranteeing sufficient structural divergence from training instances. Additional visualizations of top alignments are provided in the appendix. Figure 3 illustrates the cases achieving the minimal RMSD and minimal Earth Mover's Distance (EMD), respectively.

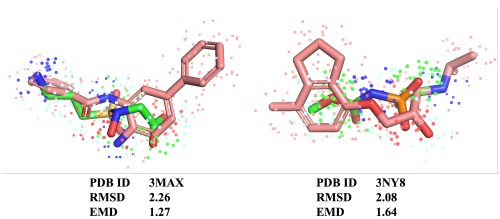

| PDB ID | 3MAX | PDB ID | 3NY8 |
| RMSD | 2.26 | RMSD | 2.08 |
| EMD | 1.27 | EMD | 1.64 |

Figure 3: **Best-aligned cases by RMSD and EMD.** The co-crystallized ligand (pink carbon skeleton, square-shaped point cloud) is aligned with a structurally similar molecule from the training set (green carbon skeleton, circular point cloud). (*Left*) alignment with min RMSD. (*Right*) alignment with min EMD.

**Baselines.** We compared our model against the most recent electron density-based generation methods: ED2Mol and ECloudGen.

For **ECloudGen**, to ensure a fair comparison, the electron density, derived from the co-crystallized ligand in the binding pocket associated with the PDB ID in DUD-E, was utilized as the conditional input. This contrasts with the methodology in the original paper, which used electron density predicted from the protein pocket structure, allowing us to isolate and compare the performance of the generative models themselves. We also evaluate its performance under the original setting. To distinguish between the two approaches, we refer to the former method as Oracle Mode.

For **ED2Mol**, we observed that the electron density generated from the pocket structure by their published model was often spatially larger than that of the co-crystallized ligand, as case shown in Figure 4. We hypothesized that this might capture a broader region of potential interactions. Therefore, to test the upper limit of their method's generative capability under this condition, we used their predicted electron density from the pocket as the condition for molecule generation.

**Evaluation.** We employed the widely adopted metrics to evaluate the performance of the generated molecules, assessing the molecules generated by these models from three perspectives: 2D drug-likeness, 3D conformation quality, and recovery of known active molecules (Liu et al., 2024).

1. **2D Drug-likeness Properties.** We assessed the overall drug-likeliness using the Quantitative Estimate of Drug-likeness (QED) score and evaluated synthetic accessibility (SAS) scores. Additionally we report the diversity of generated molecules per target.
2. **3D Conformation Quality.** We estimated the binding affinity using Glide (Friesner et al., 2004) in two distinct modes to comprehensively evaluate the predicted poses:
   - **Minimize-in-place:** The score after energy minimization of the generated pose.
   - **Redock:** The score after completely re-docking the molecule into the binding pocket.

Furthermore, we report **docking score improvement** - the ratio of molecules for which the *minimize-in-place* score is lower (i.e., better) than the *redock* score (Min-In-Place < Redock). This ratio serves as a key metric for evaluating the model's ability to generate more stable poses than those found through a standard docking search. We also report **strain energy** calculated by Posecheck, which can reveal conformation stability.

3. **Recovery of Known Active Molecules.** A generated molecule is considered to have recovered a known active molecule if its ECFP4 Tanimoto similarity to any active ligand for a given target exceeds a threshold of 0.5 (ECFP4_TS > 0.5). For a target protein to be considered successfully recovered, at least one generated molecule must meet this criterion for any of its known active ligands.

**Model Setup.** We evaluate generation under two atom-count settings:

1. **Oracle Mode**: The number of heavy atoms is set to match the co-crystallized ligand.
2. **Soft Mode**: The heavy atom count $N$ is sampled from a volume-conditioned, bounded normal distribution. Specifically, we precompute a lookup table (Table 2) from the training set: for point cloud volumes binned into 50 intervals, we fit a truncated normal distribution $\mathcal{N}_{[a,b]}(\mu_V, \sigma_V^2)$ per bin, where $\mu_V$ and $\sigma_V$ are the empirical mean and standard deviation of atom counts in that volume bin, and bounds are set to $[\mu_V - 1.5\sigma_V, \mu_V + 1.5\sigma_V]$ to avoid chemically implausible extremes. At inference, given $V_{\mathcal{P}}$, we sample $N$ from the corresponding bin's distribution.

Figure 4: **Comparison of reference and generated ED by ED2Mol.** (PDB ID: 1ZWS) The pink mesh represents the reference electron density; the grey mesh shows the generated density.

Further implementation details, including architectural hyper-parameters, training schedules, and sampling protocols, are provided in Appendix A.

## 5.2 RESULTS

Our main results are listed below:

1. Our model achieves the highest recovery rate, indicating superior potential for identifying active compounds in real-world screening.
2. It yields the greatest docking score improvement, demonstrating the ability to generate conformations that surpass those identified by standard docking methods.
3. The model generates larger molecules with generally worse QED and SAS scores than those of ED2Mol and ECloudGen, reflecting reduced drug-likeness. On molecules above 400 Da, it performs similarly to ED2Mol but remains inferior to ECloudGen.

**Ligand-Pocket Binding Pose Quality.** As shown in Table 1, we evaluated model performance using two key metrics: *min-in-place scores* and the **better min-in-place ratio**. For the absolute measure, which evaluate the quality of pocket-binding by using physical force-field, ED-BFN demonstrably outperforms baseline models, achieving the lowest (superior) docking scores across both minimize-in-place and redock assays. Regarding the relative metric, which tells the ratio of generated conformations better than those from classical force-field sampling, ED-BFN achieves such a ratio of over 30%, a performance significantly exceeding all baselines. Collectively, these results establish ED-BFN's capacity to generate ligands with markedly improved pocket-binding modes, robustly supported by both absolute and relative evaluations. We attribute this to our interaction-aware graph conditioning, which explicitly encourages geometric and electrostatic complementarity between the generated ligand and the electron density field. We further compare ED-BFN against established structure-based drug design baselines that condition solely on protein pocket geometry. Results in Appendix Table 5 demonstrate that ED-BFN maintains state-of-the-art performance.

Table 1: Comparative summary of key properties for reference and generated molecules. (↑) / (↓) indicates that higher / lower values are preferred; "–" denotes not applicable. The top two performing methods are highlighted in **bold** and underlined text, respectively.

*Abbreviations:* SE = Strain Energy; Div = Diversity; Avg = Average; Med = Median.

| Metric (Unit / Note) | Reference | Ours - O[#] | Ours | ED2Mol | ECloudGen - O[#] | ECloudGen |
|---|---|---|---|---|---|---|
| *Binding Pose Quality — All Molecules* | | | | | | |
| Min-In-Place Avg (↓) | - | **-7.21** | -6.95 | -5.24 | - | - |
| Min-In-Place Med (↓) | - | **-7.28** | -7.08 | -5.23 | - | - |
| Redock Avg (↓) | -7.93 | **-6.94** | -6.54 | -6.16 | - | - |
| Redock Med (↓) | -7.86 | **-6.82** | -6.42 | -6.15 | - | - |
| Min-In-Place < Redock (%) (↑) | - | **45.7** | 32.5 | 7.4 | - | - |
| *Binding Pose Quality — Drug-like Molecules Only* | | | | | | |
| Min-In-Place Avg (↓) | - | **-7.25** | -7.22 | -5.22 | - | - |
| Min-In-Place Med (↓) | - | -7.35 | **-7.36** | -5.24 | - | - |
| Redock Avg (↓) | -7.93* | **-7.15** | -6.97 | -6.15 | - | - |
| Redock Med (↓) | -7.86* | **-7.00** | -6.91 | -6.15 | - | - |
| Min-In-Place < Redock (%) (↑) | - | **44.2** | 36.9 | 7.3 | - | - |
| *Recovery of Bioactive Molecules* | | | | | | |
| ECFP4-TS > 0.5 (↑) | 101/101 | **37/101** | 28/101 | 3/101 | 33/101 | 6/101 |
| *Conformational Stability - All Molecules †— Strain Energy (SE, ↓)* | | | | | | |
| SE 25% | - | 44 | 55 | **14** | - | - |
| SE 50% | - | 63 | 79 | **21** | - | - |
| SE 75% | - | 88 | 117 | **31** | - | - |
| *Conformational Stability - Drug-like Molecules Only †— Strain Energy (SE, ↓)* | | | | | | |
| SE 25% | - | 43 | 44 | **14** | - | - |
| SE 50% | - | 61 | 62 | **21** | - | - |
| SE 75% | - | 86 | 91 | **31** | - | - |
| *Drug-like Properties* | | | | | | |
| SAS Avg (↓) | 3.6 | 4.5 | 4.8 | 3.9 | **2.9** | **2.9** |
| QED Avg (↑) | 0.46 | 0.49 | 0.51 | 0.73 | **0.73** | 0.66 |
| Div Avg (↑) | 0.77 | 0.74 | 0.81 | **0.89** | 0.78 | 0.87 |
| Mol. Weight | 438 | 404 | 434 | 234 | 326 | 213 |

[#] O means oracle mode.

* Reference molecules are experimentally validated bioactive ligands; thus, no additional drug-likeness filtering was applied. All docking-related metrics for the reference set are computed over the full set of active compounds in both sections.

†All methods evaluated on *min-in-place* (energy-minimized) conformations to ensure fair comparison with ED2Mol, whose pipeline inherently includes QScore-guided placement and `smina` minimization.

**Conformational Stability.** We assess the conformational strain energy of molecules generated by ED-BFN and ED2Mol (ECloudGen is excluded due to its lack of 3D output). Although ED2Mol exhibits lower strain energies in our direct comparison, this may reflect methodological biases—specifically, its use of QScore which encodes molecular electrostatic potential to guide fragment placement, and `smina` (Koes et al., 2013), a fork of AutoDock Vina optimized for scoring and energy minimization, both of which strongly penalize deviations from ideal geometries. To contextualize our results, we compare ED-BFN's strain energy distribution against the large-scale baseline from MolCRAFT. As shown in Table 4, ED-BFN's strain energies are comparable to those of MolCRAFT and fall within the typical range for drug-like molecules, indicating that its generated conformations are structurally plausible and energetically reasonable.

**Recovery of Bioactive Molecules.** ED-BFN achieves the highest recovery rate in Oracle Mode (37/101), outperforming ECloudGen (33/101) under the same condition (electron density of co-crystallized ligand). In Soft Mode, where atom count is predicted from density volume, ED-BFN recovers 28/101 targets, while ECloudGen's recovery rate under its full generation pipeline (from protein pocket to SMILES) is 6/101. ED2Mol, using its model-predicted density, recovers only 3/101 targets. These results demonstrate that ED-BFN's 3D point-cloud conditioning enables robust bioactive recovery, even without atom counts of co-crystallized ligand.

**Drug-like Properties.** As shown in Table 1, ED-BFN generates significantly larger molecules with overall worse QED and SAS scores. Figure 5 reveals that ED2Mol outperforms ED-BFN in the

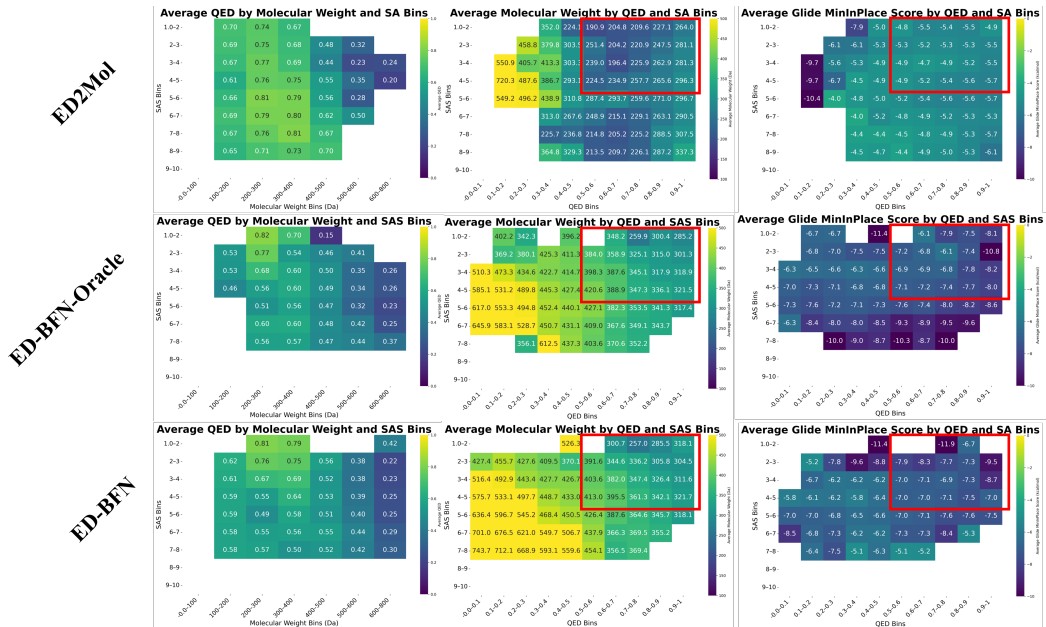

Figure 5: **Comparison of molecular property distributions between ED2Mol and ED-BFN**, visualized via heatmaps over quantitative estimate of drug-likeness (QED) and synthetic accessibility score (SAS). The red bounding boxes highlight the drug-like region (QED > 0.5 and SAS < 5).

low-molecular-weight regime, yielding smaller, better scoring ligands. Beyond 400 Da, however, its QED–SAS distribution converges with ED-BFN's, indicating its advantage is size-limited, and insufficient modeling of ligand–pocket interactions. In contrast, ECloudGen consistently achieves superior QED and SAS scores across all molecular weight ranges (Figure 8). Notably, even its rare outputs exceeding 400 Da retain high drug-like quality without degradation, indicating an intrinsic bias toward drug-like chemical space that is robust to molecular size.

## 6 DISCUSSION

Electron density provides a more comprehensive and spatially continuous description of molecular environments than discrete atomic coordinates, capturing subtle variations in shape, charge distribution, and steric occupancy that are critical for guiding biologically realistic ligand placement. While our method shares similarities with ligand-based drug design (LBDD), it is not merely a supplement to structure-based or ligand-based approaches; rather, it represents a fundamental shift in modeling philosophy—moving away from fixed atomic templates toward continuous, physics-aware representations that align more closely with experimental observations. A detailed theoretical discussion of the benefits of electron density representations, along with an assessment of current methodological limitations, can be found in Appendix Section D, where we aim to unify existing molecular representations under the electron density framework. Notably, this representation offers broad applicability in drug design, as electron density can be constructed not only from experimentally solved pocket–ligand complexes but also from ligand conformations sampled via molecular dynamics simulations. Moreover, by using low-resolution (3.5 Å) electron density of reference ligands, our approach balances geometric guidance with generative diversity while remaining robust to moderate inaccuracies in simulated structures.

We have further introduced experimental electron density data into the framework and have completed fine-tuning and testing on this data. The results demonstrate an acceptable level of active molecule recovery rate (21/101). As shown in Figure 6, we observe a clear positive correlation between the recovery rates of molecules generated using experimental electron density and those using computed electron density. Notably, in several cases, the experimental electron density-based approach yields superior recovery rates compared to the computed electron density-based method.

Specific data and analysis are available in Appendix Section A, validating our framework's capability to effectively integrate experimental and computational information.

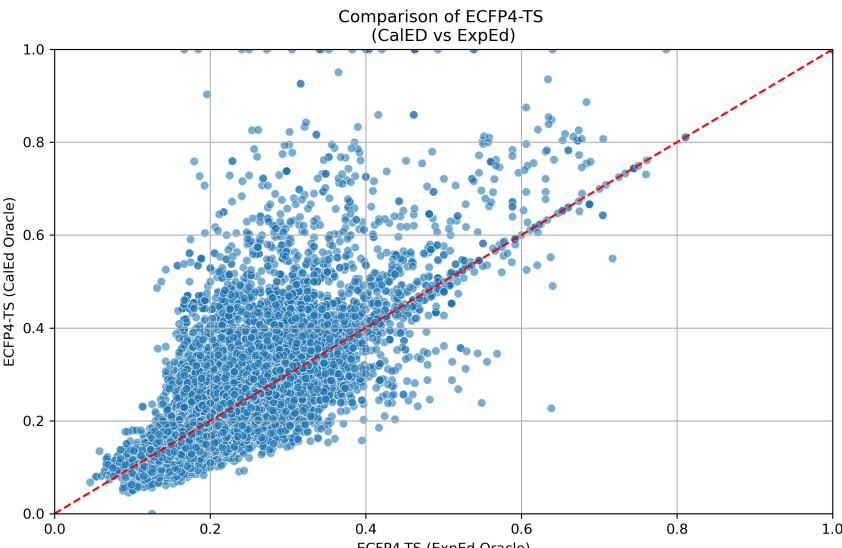

Figure 6: **Scatter plot of ECFP4-Tanimoto similarity for Calculated-ED-based and Experimental-ED-based ED-BFN.** The x-axis represents the maximum ECFP4-TS between a known active molecule and all molecules generated by the Experimental-ED-based model for a given target; the y-axis represents the corresponding value for the Calculated-ED-based model. Each point corresponds to one active molecule for a specific target.

Regarding the method for obtaining electron density, our use of the CCTBX toolkit presents clear advantages over alternative quantum chemical approaches. As shown in the distribution comparison in Appendix Figure 10, electron densities derived from traditional quantum chemical methods (here computed using Multiwfn (Lu & Chen, 2012; Lu, 2024)) exhibit larger deviations from experimental observations. In contrast, CCTBX employs empirical crystallographic data and refinement techniques to produce electron density distributions that align more closely with experimental results. This alignment ensures that our modeling framework is grounded in physical reality, thereby offering a more reliable foundation for drug design.

Future work could focus on developing multi-resolution conditioning schemes, extracting informative geometric and topological features from the density field $\rho_P$, and adapting the framework to better incorporate experimental structural data.

## 7 CONCLUSION

ED-BFN demonstrates that sparse, pharmacophore-annotated electron density point clouds serve as effective structural priors for 3D molecular generation. By conditioning GeoBFN through a dynamically constructed interaction graph, our method generates ligands with improved geometric fidelity and binding affinity, without requiring explicit pocket definitions or voxelized grids. This enables robust ligand design in the presence of partial or ambiguous structural information and establishes a general framework for conditioning generative models on continuous, non-atomic structural observables. We anticipate that this representation will enable generative models to better approximate the noisy, dynamic reality of structural biology, moving beyond static, idealized atomic coordinates toward experimentally faithful molecular design.

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

SUPPLEMENTARY MATERIAL

OVERVIEW

This appendix provides comprehensive experimental and evaluation details, additional visualizations, and theoretical justification for using electron density as a structural representation. We present comparative results against pocket-based SBDD methods, in-depth analyses of molecules generated by our ED-based approach, and a theoretical discussion on how electron density can abstract diverse conditioning signals for molecular generation. We also examine what structural information is inherently lost in our point cloud representation and in other representations more generally. The content is organized into four main sections:

1. Experimental Detail
2. Additional Experimental Results
3. Pseudocode For ED-BFN Generation
4. From Schrödinger To Pharmacophore: Why We Choose Electron Density As Our Representations

## A    EXPERIMENTAL DETAIL

We build upon the **GeoBFN** framework as implemented in **MolCRAFT**, retaining all default architectural and training hyperparameters unless otherwise specified. The key architectural modification lies in replacing the original *pocket encoder* with an *electron density encoder*, achieved by adjusting the input dimension to **4**. The lookup table for soft mode can be found at Table 2.

**Electron Density Representation**    Electron density grids are computed using **cctbx** Grosse-Kunstleve et al. (2002) as described in the main text, with resolution set to **3.5 Å** and voxel grid spacing of **0.5 Å**. After computing the full 3D electron density field $\rho(\mathbf{r})$, we retain only grid points where $\rho > 0.6$ to focus on chemically relevant regions. If the number of selected points exceeds our maximum limit of **200**, we apply uniform downsampling to meet this constraint.

**Model Architecture and Training**    We use the default GeoBFN configuration from MolCRAFT: **9 layers**, **128 hidden dimensions**, **16 heads**. Training uses the **Muon optimizer** (Kimi K2 implementation (Liu et al., 2025)) with learning rate `5e-4`, weight decay `0.01`, and batch size **8**. The learning rate scheduler is `ReduceLROnPlateau` (mode=`'min'`, factor=`0.6`, patience=`10`, min_lr=`1e-6`). Validation is performed every **10,000 epochs** on a random sample of **10,000 examples** from the held-out validation set (drawn from the remaining ∼400M of the 2.4B dataset). Training runs for **40 hours** on a single NVIDIA RTX 4090 (24GB VRAM), and the checkpoint at epoch **310,000** is selected for final evaluation based on optimal validation performance.

**Generation Protocol for ECloudGen**    At inference time, molecule generation proceeds in two stages:

**Pocket → Filled Cloud**: Given the pocket electron density, we generate **8 samples** of filled electron clouds.

**Filled Cloud → SMILES**: Each filled cloud is decoded into **1,000 molecular SMILES strings** via our autoregressive decoder.

Final molecules are filtered for validity and deduplicated before evaluation.

**Evaluation Metrics**    We report the following standard metrics for generated molecules:

- **Docking Score**: Computed using **Schrödinger Glide** (SP mode, default settings) to assess binding affinity.
- **QED** (Quantitative Estimate of Drug-likeness) and **SAS** (Synthetic Accessibility Score): Calculated via `RDKit` APIs.

Table 2: Lookup Table for Soft Mode: Volume ranges and corresponding atomic statistics

| Volume Ranges [Low, High) | $\mu$ | $\sigma$ | Volume Ranges [Low, High) | $\mu$ | $\sigma$ |
|---|---|---|---|---|---|
| [43.27, 174.72) | 15.14 | 2.43 | [347.77, 352.63) | 31.80 | 2.97 |
| [174.72, 196.11) | 17.97 | 2.27 | [352.63, 357.50) | 32.14 | 3.01 |
| [196.11, 211.74) | 19.61 | 2.21 | [357.50, 362.52) | 32.48 | 3.05 |
| [211.74, 224.18) | 20.89 | 2.19 | [362.52, 367.62) | 32.79 | 3.08 |
| [224.18, 234.62) | 21.95 | 2.19 | [367.62, 372.86) | 33.16 | 3.12 |
| [234.62, 243.62) | 22.85 | 2.23 | [372.86, 378.23) | 33.52 | 3.17 |
| [243.62, 251.69) | 23.61 | 2.29 | [378.23, 383.73) | 33.84 | 3.22 |
| [251.69, 258.98) | 24.30 | 2.31 | [383.73, 389.45) | 34.18 | 3.26 |
| [258.98, 265.72) | 24.94 | 2.34 | [389.45, 395.35) | 34.54 | 3.31 |
| [265.72, 272.07) | 25.48 | 2.37 | [395.35, 401.51) | 34.88 | 3.38 |
| [272.07, 278.07) | 26.02 | 2.41 | [401.51, 408.02) | 35.27 | 3.44 |
| [278.07, 283.80) | 26.51 | 2.46 | [408.02, 414.96) | 35.71 | 3.50 |
| [283.80, 289.29) | 26.98 | 2.49 | [414.96, 422.26) | 36.15 | 3.58 |
| [289.29, 294.62) | 27.42 | 2.52 | [422.26, 430.08) | 36.58 | 3.66 |
| [294.62, 299.77) | 27.84 | 2.56 | [430.08, 438.42) | 37.05 | 3.73 |
| [299.77, 304.79) | 28.26 | 2.60 | [438.42, 447.52) | 37.59 | 3.83 |
| [304.79, 309.73) | 28.67 | 2.63 | [447.52, 457.56) | 38.15 | 3.94 |
| [309.73, 314.56) | 29.02 | 2.68 | [457.56, 468.68) | 38.72 | 4.05 |
| [314.56, 319.32) | 29.39 | 2.70 | [468.68, 481.35) | 39.45 | 4.20 |
| [319.32, 324.09) | 29.75 | 2.74 | [481.35, 495.89) | 40.28 | 4.35 |
| [324.09, 328.78) | 30.12 | 2.78 | [495.89, 513.21) | 41.21 | 4.56 |
| [328.78, 333.46) | 30.46 | 2.82 | [513.21, 534.86) | 42.37 | 4.81 |
| [333.46, 338.21) | 30.80 | 2.86 | [534.86, 564.34) | 43.83 | 5.08 |
| [338.21, 342.95) | 31.12 | 2.90 | [564.34, 612.42) | 45.71 | 5.34 |
| [342.95, 347.77) | 31.50 | 2.93 | [612.42, $+\infty$) | 48.77 | 5.34 |

# B  ADDITIONAL EXPERIMENTAL RESULTS

## B.1  CHECK INFORMATION LEAKAGE BETWEEN TRAINING DATA AND DUD-E

To evaluate potential structural overlap between training molecules and DUD-E reference binding pockets, we compute the minimum Earth Mover's Distance (EMD) following optimal global rigid-body alignment. All alignment results for reconstructed bioactive conformations are summarized in Table 3.

Table 3: RMSD and EMD values for optimal alignments between training molecules and DUD-E reference pockets

| PDB ID | RMSE | EMD | PDB ID | RMSE | EMD | PDB ID | RMSE | EMD | PDB ID | RMSE | EMD |
|--------|------|------|--------|------|------|--------|------|------|--------|------|------|
| 3g0e | 2.99 | 2.03 | 2oi0 | 2.37 | 1.50 | 3hmm | 2.35 | 1.66 | 3lpb | 2.67 | 1.90 |
| 2qd9 | 3.00 | 3.89 | 2zdt | 2.85 | 2.20 | 3krj | 3.00 | 2.70 | 1lru | 2.74 | 1.72 |
| 830c | 2.61 | 1.52 | 2znp | 3.00 | 2.70 | 3kgc | 2.78 | 1.77 | 2e1w | 2.44 | 2.40 |
| 3ccw | 3.00 | 2.79 | 3lq8 | 2.96 | 1.96 | 1sqt | 2.37 | 1.56 | 1b9v | 2.59 | 2.28 |
| 3chp | 2.70 | 2.56 | 3l3m | 2.41 | 1.41 | 1sj0 | 3.00 | 2.36 | 2b8t | 2.31 | 1.75 |
| 2oyu | 3.00 | 2.25 | 3biz | 3.00 | 3.53 | 3kba | 2.71 | 1.67 | | | |
| 3max | 2.26 | 1.27 | 3ny8 | 2.08 | 1.64 | 1q4x | 3.00 | 2.12 | | | |
| 1w7x | 3.00 | 2.58 | 3eqh | 2.32 | 2.29 | 3bz3 | 3.00 | 2.46 | | | |
| 2i78 | 2.13 | 1.60 | 3kl6 | 2.97 | 1.89 | 3l5d | 2.61 | 1.38 | | | |
| 2am9 | 2.58 | 2.14 | 2p2i | 3.00 | 3.40 | 2fsz | 2.57 | 2.84 | | | |

For each training molecule $\mathcal{L} = (\mathbf{X}, \mathbf{z})$ and each reference molecule $\mathcal{L}_i^{\text{ref}} = (\mathbf{X}_i^{\text{ref}}, \mathbf{z}_i^{\text{ref}})$, we:

1. Sample corresponding points per atom type;
2. Fit a global rigid transform $(\mathbf{R}, \mathbf{t})$ via Kabsch algorithm;
3. Apply transform to entire source point cloud;
4. Compute EMD between transformed source and target.

The full procedure is formalized in Algorithm 1.

## B.2  COMPARISON ON MIN-IN-PLACE DOCKING SCORE WITH ED2MOL ON DIFFERENT MOLECULAR WEIGHTS

Considering that ED2Mol utilizes the Ligand Efficiency (LE) metric, which accounts for the influence of molecular size on docking scores, we directly present the distribution of min-in-place docking scores across different molecular weights. The results indicate that for all generated molecules, ED2Mol only fails to exhibit lower docking scores than our method in the range above 500 Da. Moreover, for drug-like molecules, ED2Mol not only shows inferior docking scores across all weight ranges but also struggles to produce drug-like molecules with weights exceeding 500 Da.

## B.3  COMPARISON ON TEST DATA USED BY MOLCRAFT

To provide a more comprehensive presentation of the data distribution, we utilized the test results published by MolCRAFT on the CrossDocked dataset. Due to time constraints, we employed oracle mode and randomly selected 20 target proteins from their test set for molecule generation. Docking was performed using Vina, following MolCRAFT's testing protocol. The Strain Energy evaluation was conducted on In-Place conformations rather than Min-In-Place conformations. The results are shown in Table 4.

## B.4  COMPARISON ON DUD-E DATA WITH OTHER SBDD METHODS

Although this work focuses on generation conditioned on electron density maps, and we did not explicitly present detailed comparison results with other SBDD methods in the main text, we still include the most critical evaluation metrics in the appendix. Specifically, we compare ED-Mol with Pocket2Mol, TargetDiff, Lingo3DMol, and MolCRAFT in terms of docking performance and bioactive molecule recovery. As shown in the Table 5, ED-Mol consistently achieves the best results.

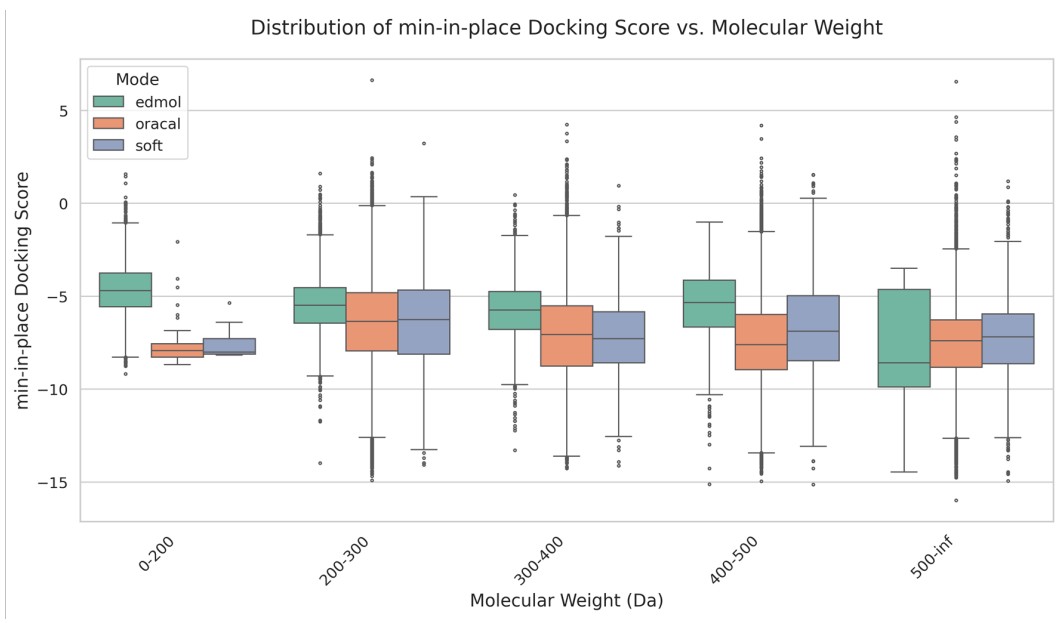

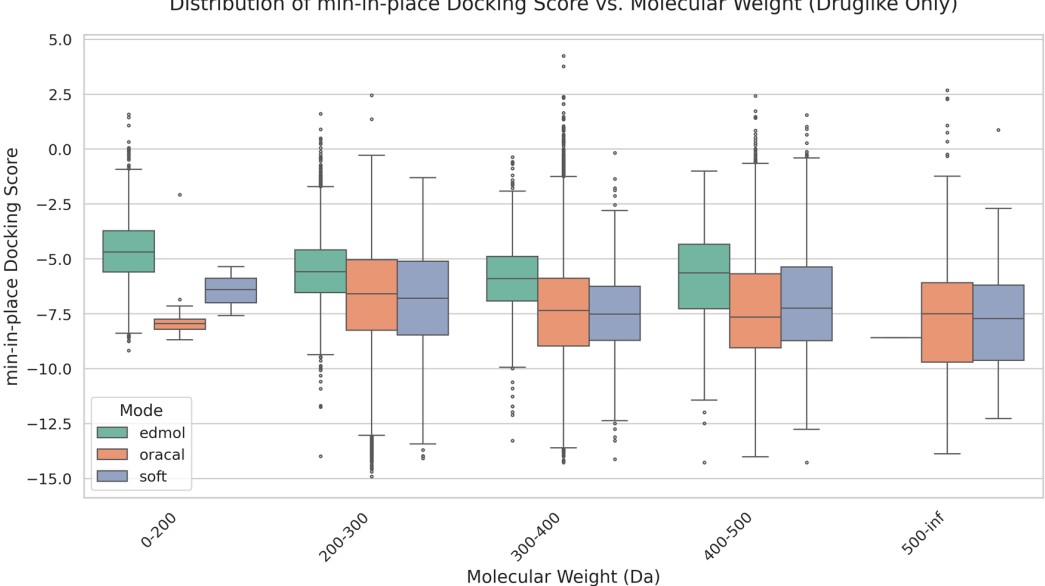

Figure 7: Comparison of min-in-place docking scores between our method and ED2Mol across different molecular weight intervals. Our approach demonstrates superior or comparable docking performance in most ranges, particularly for drug-like molecules above 500 Da where ED2Mol exhibits limitations.

---

**Algorithm 1** Global Rigid Alignment and EMD Calculation

---

**Require:** Source molecule $(\mathbf{X}, \mathbf{z})$, target molecule $(\mathbf{X}^{\text{ref}}, \mathbf{z}^{\text{ref}})$, both with 3D coordinates and atom type labels.

**Ensure:** Transformed source points $\mathbf{X}_{\text{aligned}}$, EMD score, rotation $\mathbf{R}$, translation $\mathbf{t}$.

1: $\mathcal{S}_{\text{src}} \leftarrow [\,], \mathcal{S}_{\text{tgt}} \leftarrow [\,]$
2: **for** each atom type $l \in \text{unique}(\mathbf{z}) \cap \text{unique}(\mathbf{z}^{\text{ref}})$ **do**
3:     $\mathbf{X}_l \leftarrow \{\mathbf{x} \in \mathbf{X} \mid z = l\}, \mathbf{X}_l^{\text{ref}} \leftarrow \{\mathbf{x} \in \mathbf{X}^{\text{ref}} \mid z = l\}$
4:     **if** $|\mathbf{X}_l| = 0$ or $|\mathbf{X}_l^{\text{ref}}| = 0$ **then**
       **continue**
5:     **end if**
6:     $n_l \leftarrow \min(|\mathbf{X}_l|, |\mathbf{X}_l^{\text{ref}}|)$
7:     **if** $n_l < 3$ **then**
       **continue**
8:     **end if**
9:     Append first $n_l$ points of $\mathbf{X}_l$ to $\mathcal{S}_{\text{src}}$
10:    Append first $n_l$ points of $\mathbf{X}_l^{\text{ref}}$ to $\mathcal{S}_{\text{tgt}}$
11: **end for**
12: **if** $|\mathcal{S}_{\text{src}}| < 3$ **then**
13:    **return** $\mathbf{X}, \mathbf{X}^{\text{ref}}$, None, None
14: **end if**
15: $\mathbf{P} \leftarrow \text{concat}(\mathcal{S}_{\text{src}}), \mathbf{Q} \leftarrow \text{concat}(\mathcal{S}_{\text{tgt}})$
16: $\boldsymbol{\mu}_P \leftarrow \text{mean}(\mathbf{P}), \boldsymbol{\mu}_Q \leftarrow \text{mean}(\mathbf{Q})$
17: $\mathbf{P}_c \leftarrow \mathbf{P} - \boldsymbol{\mu}_P, \mathbf{Q}_c \leftarrow \mathbf{Q} - \boldsymbol{\mu}_Q$
18: $\mathbf{H} \leftarrow \mathbf{P}_c^\top \mathbf{Q}_c$
19: $\mathbf{U}, \_, \mathbf{V}^\top \leftarrow \text{SVD}(\mathbf{H})$
20: $\mathbf{R} \leftarrow \mathbf{V}\mathbf{U}^\top$
21: **if** $\det(\mathbf{R}) < 0$ **then**
22:    $\mathbf{V}[:, -1] \leftarrow -\mathbf{V}[:, -1]$
23:    $\mathbf{R} \leftarrow \mathbf{V}\mathbf{U}^\top$
24: **end if**
25: $\mathbf{t} \leftarrow \boldsymbol{\mu}_Q - \mathbf{R}\boldsymbol{\mu}_P$
26: $\mathbf{X}_{\text{centered}} \leftarrow \mathbf{X} - \boldsymbol{\mu}_P$
27: $\mathbf{X}_{\text{aligned}} \leftarrow (\mathbf{R}\mathbf{X}_{\text{centered}}^\top)^\top + \boldsymbol{\mu}_Q$
28: $\mathbf{P}_{\text{aligned}} \leftarrow (\mathbf{R}\mathbf{P}_c^\top)^\top + \boldsymbol{\mu}_Q$
29: $\text{emd} \leftarrow EMD(\mathbf{P}_{\text{aligned}}, \mathbf{Q})$
30: **return** $\mathbf{X}_{\text{aligned}}, \text{emd}, \mathbf{R}, \mathbf{t}$

---

**Algorithm 2** Earth Mover's Distance (EMD) via Linear Assignment

---

**Require:** Two point sets $\mathbf{A} \in \mathbb{R}^{n \times 3}, \mathbf{B} \in \mathbb{R}^{m \times 3}$

**Ensure:** EMD = average minimal matching cost

1: **if** $n = 0$ or $m = 0$ **then**
2:    **return** $+\infty$
3: **end if**
4: $\mathbf{C} \leftarrow \text{pairwise\_euclidean\_distance}(\mathbf{A}, \mathbf{B}) \ \{\mathbf{C}_{ij} = \|\mathbf{a}_i - \mathbf{b}_j\|_2\}$
5: $(\text{row\_ind}, \text{col\_ind}) \leftarrow \text{linear\_sum\_assignment}(\mathbf{C})$
6: $\text{total\_cost} \leftarrow \sum_k \mathbf{C}[\text{row\_ind}_k, \text{col\_ind}_k]$
7: $\text{emd} \leftarrow \text{total\_cost} / \min(n, m) \ \{\text{Normalize by number of matched pairs}\}$
8: **return** emd

---

## B.5 MORE DETAILS FOR DRUG-LIKE PROPERTIES

To provide a holistic view of performance across energy-based molecular generation approaches, we present in Figure 8 a comparative heatmap analysis of all ED-based methods evaluated in this work. The distributions over QED and SAS reveal that our proposed method achieves higher concentration within the drug-like region (QED < 0.5, SAS < 5), indicating superior pharmaceutical relevance. For completeness, we additionally include results from *Counts*-based sampling — a

Table 4: Comparison of molecular docking scores, drug-likeness, and strain across methods on MolCRAFT Published Data. Vina scores (↓ = lower better), QED (↑ = higher better), SA (↓ = lower better), Strain (25%, 50%, 75% quantiles, ↓ = lower better).

| Methods | Vina_score | | Vina_minimize | | Vina_dock | | QED | SA | Strain (kcal/mol) | | |
|---|---|---|---|---|---|---|---|---|---|---|---|
| | Avg. | Med. | Avg. | Med. | Avg. | Med. | | | 25% | 50% | 75% |
| Reference | -6.93 | -6.79 | -6.88 | -6.63 | -7.49 | -7.73 | 0.47 | 3.20 | 31.94 | 106.88 | 274.83 |
| DecompDiff | -5.61 | -5.52 | -6.27 | -6.07 | -7.23 | -7.06 | 0.51 | 4.15 | 204.04 | 551.01 | 1446.83 |
| TargetDiff | -6.16 | -6.30 | -6.83 | -6.87 | -7.78 | -7.83 | 0.50 | 4.70 | 368.89 | 1195.40 | 10931.92 |
| IPDiff | -6.79 | -6.69 | -7.62 | -7.27 | -8.73 | -8.27 | 0.51 | 4.69 | 419.06 | 4605.34 | 3131179.43 |
| AR | -5.97 | -5.88 | -6.28 | -6.17 | -6.90 | -6.86 | 0.52 | 4.19 | 295.79 | 700.97 | 2507.88 |
| FLAG | 49.40 | 39.20 | 7.36 | -2.80 | -5.72 | -5.76 | 0.61 | 4.33 | 24.63 | 75.38 | 4270.96 |
| Pkt2Mol | -5.86 | -5.62 | -7.06 | -6.76 | -7.64 | -7.53 | 0.59 | 3.18 | 121.46 | 242.39 | 448.24 |
| binddm | -6.81 | -6.88 | -7.62 | -7.37 | -8.45 | -8.27 | 0.52 | 4.15 | 595.20 | 11365.56 | 8223743.15 |
| MOLCRAFT | -7.67 | -7.68 | -7.85 | -7.81 | -8.35 | -8.38 | 0.51 | 3.81 | 81.25 | 205.49 | 589.04 |
| Ours(Oracle) | -5.45 | -5.45 | -6.04 | -6.14 | -6.92 | -6.98 | 0.45 | 4.06 | 106.80 | 206.10 | 592.80 |

Table 5: Comparative summary of key properties for reference and generated molecules. "–" denotes not applicable. The top two performing methods are highlighted in **bold** and underlined text, respectively.

| Method / Metric | Min-In-Place Avg | Redock Avg | Min-In-Place < Redock (%) | ECFP4-TS > 0.5 |
|---|---|---|---|---|
| Reference | - | -7.9 | - | - |
| Ours - Oracle | **-7.2** | -6.9 | **45.7** | **37/101** |
| Ours | -5.2 | -6.5 | 32.5 | 28/101 |
| Pocket2Mol | -6.7 | -7.5 | 17.9 | 8/101 |
| TargetDiff | -6.2 | -7.0 | 15.2 | 3/101 |
| Lingo3DMol | -6.8 | **-7.8** | 12.0 | 33/101 |
| MolCRAFT | -6.1 | -6.9 | 20.1 | 17/101 |

simple yet competitive baseline — and both generation modes of ECloudGen, demonstrating the robustness and flexibility of our framework under varied sampling strategies.

### B.6 ALIGNMENT RESULTS BETWEEN EXPERIMENTAL AND CALCULATED ELECTRON DENSITIES (ED)

We present the alignment results in the table below. Due to the stochastic nature of the Iterative Closest Point (ICP) algorithm, we performed five independent runs and report the corresponding statistical summary (e.g., mean ± standard deviation).

### B.7 PRELIMINARY INVESTIGATION OF ELECTRON DENSITY FROM MULTI-RESOLUTION CALCULATIONS

We generated molecular structures using reference structure 4TS0 at three different resolutions: 1.5Å, 3.5Å, and 8.0Å. As shown in Figure 9, the resulting molecules were subsequently fragmented using the BRICS method, and the resulting fragments were statistically analyzed. The results clearly reveal notable differences in the molecular structures generated by the current model across the various resolutions.

### B.8 MORE DETAILS FOR ACTIVE MOLECULE RECOVERY AND PRELIMINARY INVESTIGATION OF EXPERIMENTAL ELECTRON DENSITY

We provide reproduction results for all 101 target models. Here, *Active* refers to the molecule from the original active set that shows the highest similarity to the reproduced molecule; *Generated* denotes the generated molecule with the highest similarity to known active molecules for the corresponding target. *Exp ED* and *Cal ED* indicate molecules generated under experimentally derived and computationally derived electron density (ED) conditions, respectively. The terms *Oracle* and

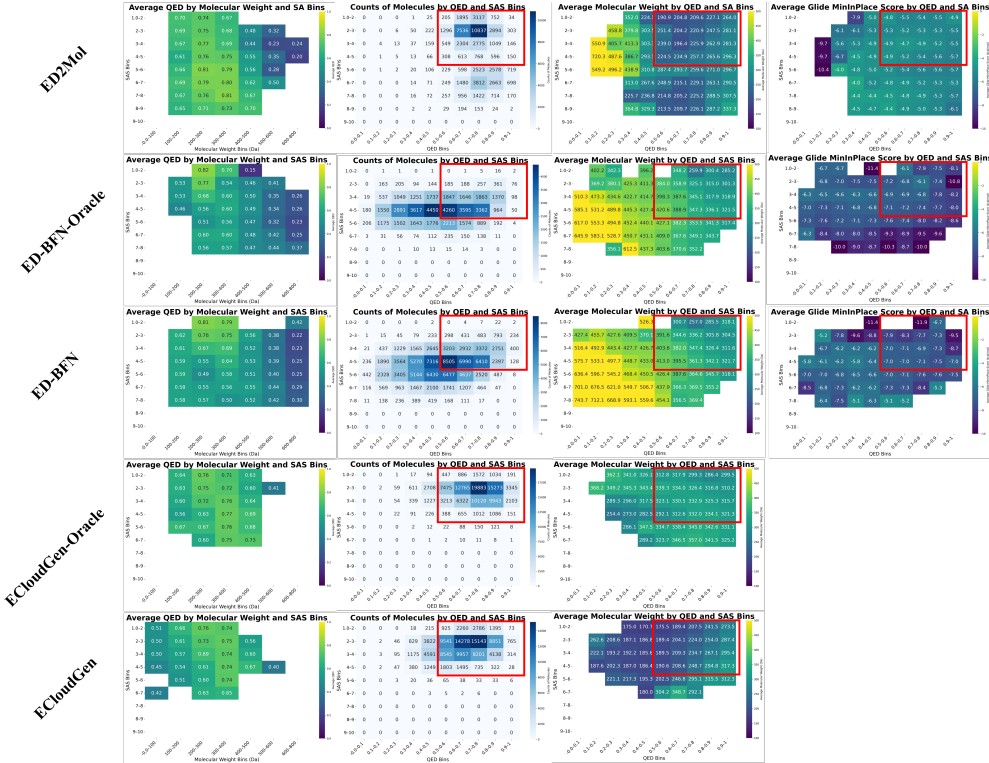

Figure 8: **Comparative analysis of molecular property distributions across all ED-based generative methods.** Heatmaps visualize density over QED and SAS, with the red bounding box indicating the drug-like region (QED > 0.5, SAS < 5). Beyond methods discussed in the main text, we include *Counts*-based sampling and two generation modes of ECloudGen for comprehensive evaluation.

*Soft* refer to two different strategies for controlling the number of atoms during generation: the former uses the ground-truth atom count (oracle-guided), while the latter applies a soft constraint (e.g., via loss regularization or probabilistic modeling).

As shown in Figure 10, the intensity distributions of CalED and ExpED differ considerably. It is worth noting that the computational electron density (CalED) was derived using two distinct methods: one based on `cctbx` and the other using `xtb + multiwfn`. The significant distributional discrepancies among these electron density types make direct transfer from CalED to ExpED challenging in our current setup. Algorithm 6 outlines the point cloud sampling procedure for ExpED.

We conducted the experiment in two stages. In the first version, we used point clouds sampled from experimental electron density, but the model itself had not been fine-tuned on experimental ED data. The results from this version were limited—in fact, only molecules generated by the oracle mode for target `2b8t` using ExpED achieved an ECFP4-TS of 0.5. This outcome highlighted the difficulty of directly transferring a CalED-pretrained model to ExpED conditions.

In the second version, we started from a model pretrained on computational electron density (CalED), fine-tuned it on experimental electron density (Exp ED), and then performed evaluation. After fine-tuning, we evaluated the model under oracle mode using Exp ED and achieved a recovery rate of $\frac{21}{101}$. We also observed that the fine-tuned model outperformed the pretrained model in several cases, as illustrated in Figure 11, 12, 13, and 14.

Table 6: Earth Mover's Distance (EMD) between point clouds sampled from experimental and calculated electron densities (ED). Units: angstrom.

| pdb_id | emd_mean | emd_std | pdb_id | emd_mean | emd_std | pdb_id | emd_mean | emd_std |
|---|---|---|---|---|---|---|---|---|
| 3lq8 | 1.17 | 0.00 | 1d3g | 1.59 | 0.02 | 3bz3 | 1.34 | 0.00 |
| 2oyu | 1.25 | 0.00 | 1q4x | 1.06 | 0.00 | 3nxu | 1.26 | 0.00 |
| 3g6z | 1.06 | 0.00 | 3lpb | 1.03 | 0.00 | 1uyg | 1.17 | 0.04 |
| 2zec | 1.28 | 0.08 | 3m2w | 1.08 | 0.00 | 1w7x | 1.32 | 0.07 |
| 2nnq | 1.37 | 0.05 | 2cnk | 2.16 | 0.07 | 3nf7 | 1.25 | 0.10 |
| 2etr | 1.05 | 0.03 | 3nxo | 1.46 | 0.13 | 3f07 | 1.50 | 0.04 |
| 3chp | 1.64 | 0.02 | 2fsz | 1.49 | 0.03 | 3el8 | 1.42 | 0.00 |
| 2rgp | 1.07 | 0.02 | 3l3m | 1.15 | 0.03 | 3eqh | 1.25 | 0.00 |
| 3l5d | 1.24 | 0.03 | 2vt4 | 0.92 | 0.01 | 3lan | 1.21 | 0.03 |
| 3kba | 1.21 | 0.03 | 2hzi | 1.18 | 0.05 | 2aa2 | 1.04 | 0.05 |
| 3d4q | 1.24 | 0.00 | 2qd9 | 1.34 | 0.00 | 2zdt | 1.06 | 0.01 |
| 3hmm | 1.45 | 0.13 | 1li4 | 1.39 | 0.08 | 3ny8 | 1.21 | 0.00 |
| 3f9m | 1.22 | 0.04 | 1s3b | 1.70 | 0.00 | 2hv5 | 1.48 | 0.08 |
| 3bgs | 1.01 | 0.00 | 1sj0 | 1.14 | 0.00 | 3e27 | 1.35 | 0.00 |
| 1zw5 | 1.01 | 0.00 | 2azr | 1.15 | 0.03 | 1kvo | 1.09 | 0.00 |
| 1ype | 1.16 | 0.05 | 3frj | 1.04 | 0.00 | 1udt | 1.12 | 0.00 |
| 3cqw | 1.15 | 0.04 | 3pbl | 1.22 | 0.00 | 1qw6 | 1.07 | 0.04 |
| 2v3f | 1.14 | 0.04 | 3biz | 1.23 | 0.00 | 1l2s | 1.11 | 0.03 |
| 2owb | 1.09 | 0.00 | 3odu | 1.15 | 0.00 | 3cjo | 1.17 | 0.00 |
| 2b8t | 1.07 | 0.02 | 3g0e | 1.66 | 0.12 | 3bwm | 1.16 | 0.04 |
| 2gtk | 0.98 | 0.00 | 3hl5 | 1.20 | 0.03 | 3max | 1.60 | 0.11 |
| 3krj | 1.17 | 0.00 | 1e66 | 1.03 | 0.05 | 3ln1 | 0.96 | 0.01 |
| 1njs | 1.26 | 0.00 | 3kl6 | 1.91 | 0.10 | 2am9 | 1.13 | 0.13 |
| 2znp | 1.13 | 0.00 | 2oj9 | 1.60 | 0.05 | 2ayw | 2.75 | 0.08 |
| 3ccw | 1.21 | 0.00 | 1b9v | 1.25 | 0.02 | 1sqt | 1.33 | 0.07 |
| 2i78 | 1.18 | 0.01 | 1vso | 1.06 | 0.01 | 1bcd | 1.02 | 0.05 |
| 2ica | 1.89 | 0.04 | 3bqd | 1.04 | 0.00 | 2ojg | 1.21 | 0.08 |
| 3d0e | 1.14 | 0.00 | 1h00 | 1.32 | 0.03 | | | |
| 1xl2 | 1.65 | 0.02 | 2e1w | 1.27 | 0.01 | | | |
| 3kgc | 1.55 | 0.08 | 2p54 | 1.32 | 0.04 | | | |
| 2p2i | 1.10 | 0.02 | 3eml | 1.48 | 0.02 | | | |

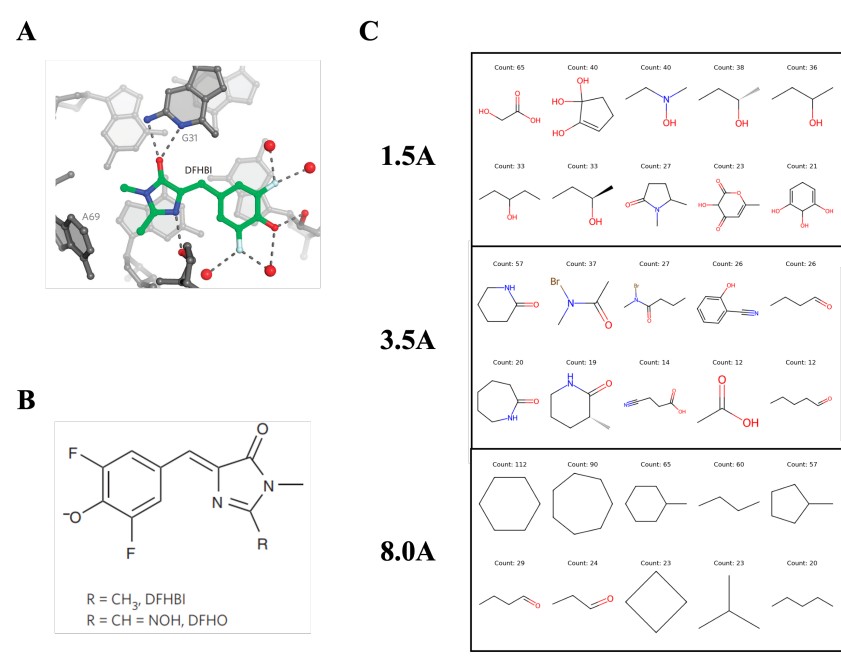

Figure 9: **Case study on 4TS0 for multi-resolution calculations.** (A) Reference structure of the ligand in 4TS0. (B) Corresponding molecular graph. (C) Distribution of fragment counts obtained via BRICS decomposition.

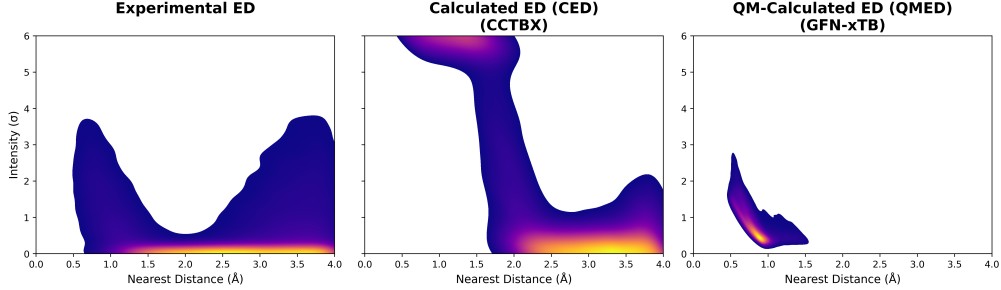

Figure 10: **Comparison of electron density intensity distributions across different calculation methods**. From left to right: (a) Experimental electron density (ExpED); (b) Computational electron density calculated using `cctbx`; (c) Computational electron density calculated using `xtb + multiwfn`.

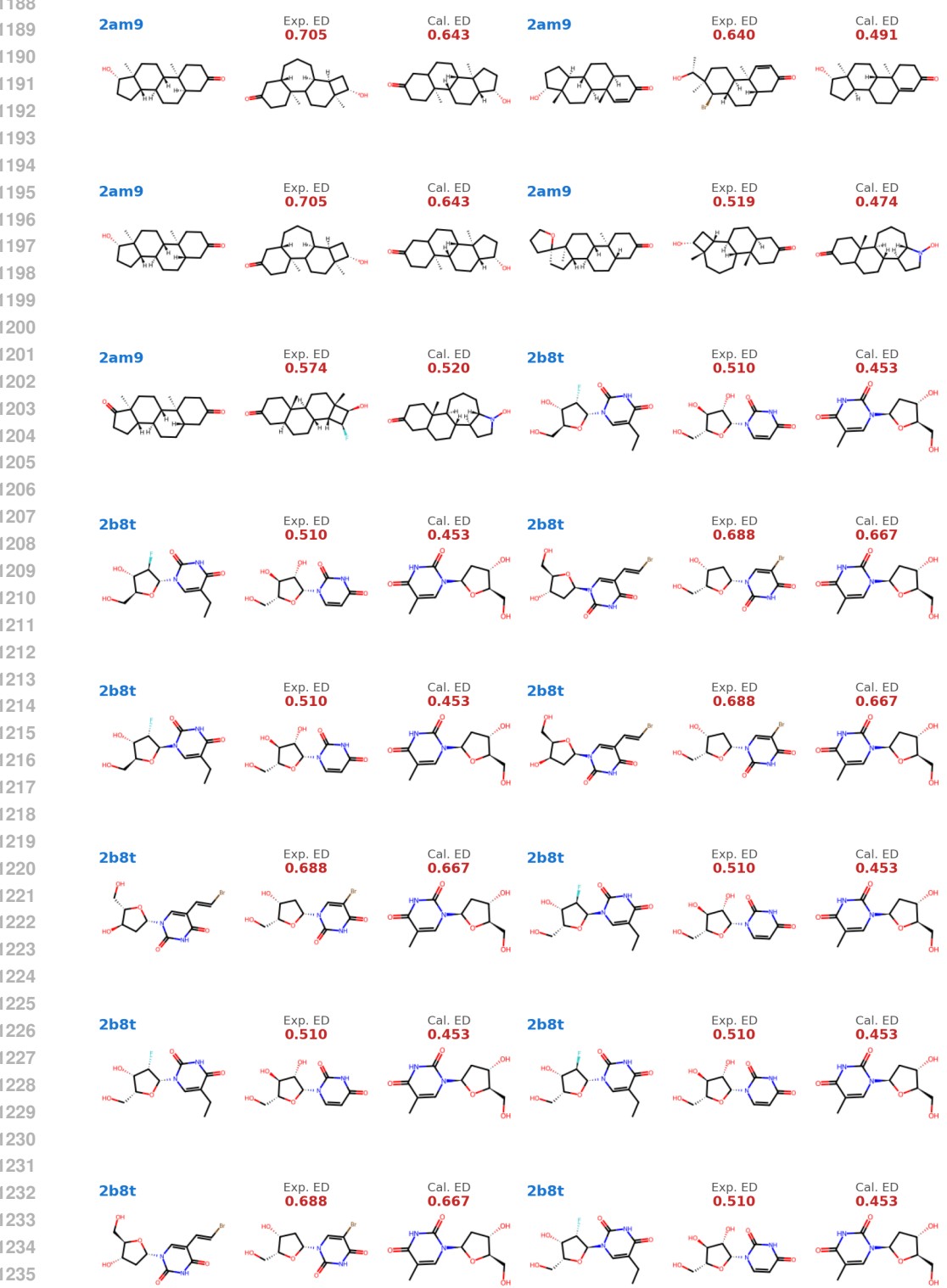

Figure 11: **Enhanced Fidelity of Molecular Recovery Using Experimental Electron Density (Part 1)**. This figure exemplifies a case where the structure refined against experimental electron density (middle) more accurately recapitulates the geometry of the active molecule (left) compared to its computationally-derived counterpart (right).

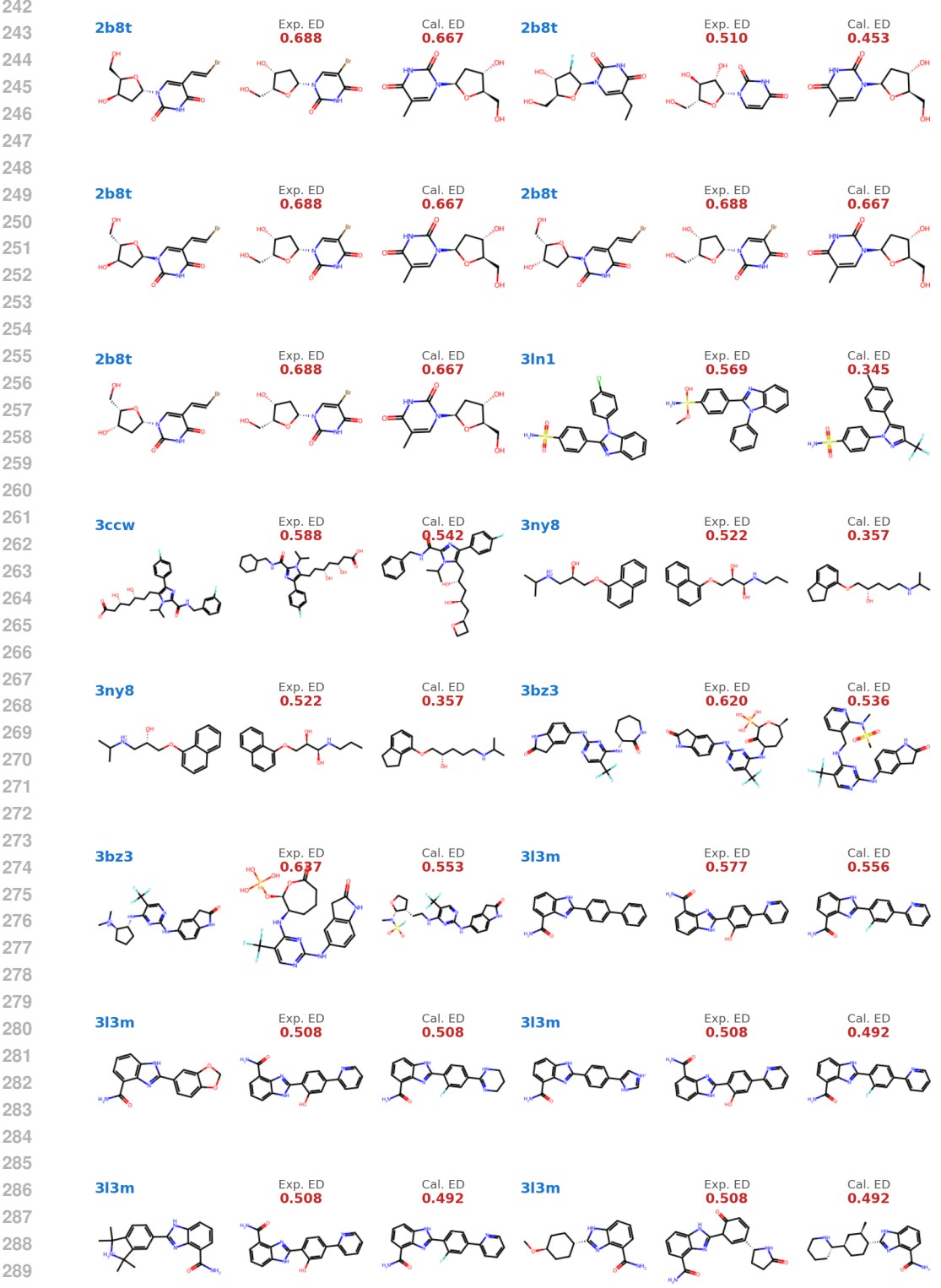

Figure 12: **Enhanced Fidelity of Molecular Recovery Using Experimental Electron Density (Part 2)**. This figure exemplifies a case where the structure refined against experimental electron density (middle) more accurately recapitulates the geometry of the active molecule (left) compared to its computationally-derived counterpart (right).

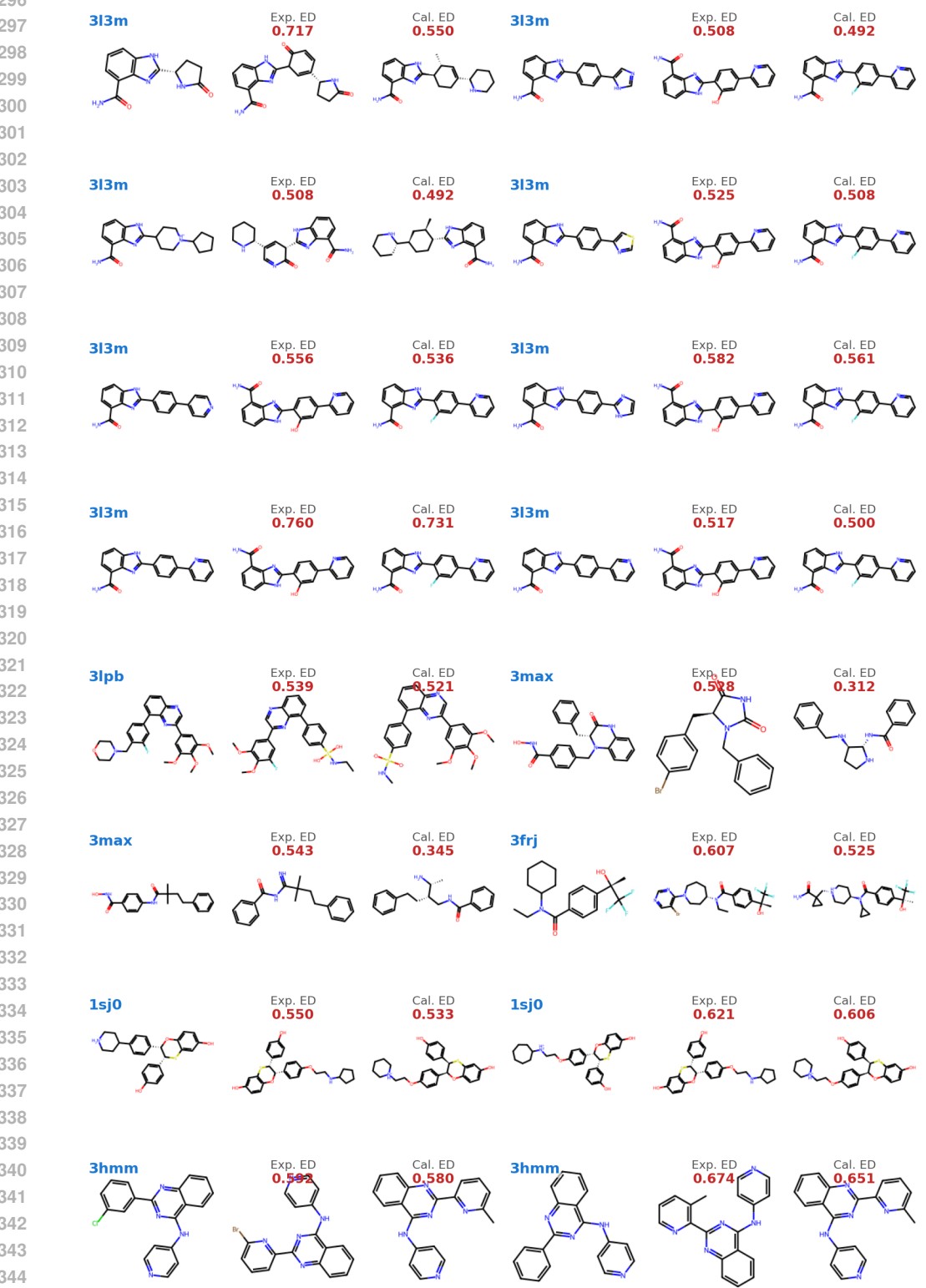

Figure 13: **Enhanced Fidelity of Molecular Recovery Using Experimental Electron Density (Part 3)**. This figure exemplifies a case where the structure refined against experimental electron density (middle) more accurately recapitulates the geometry of the active molecule (left) compared to its computationally-derived counterpart (right).

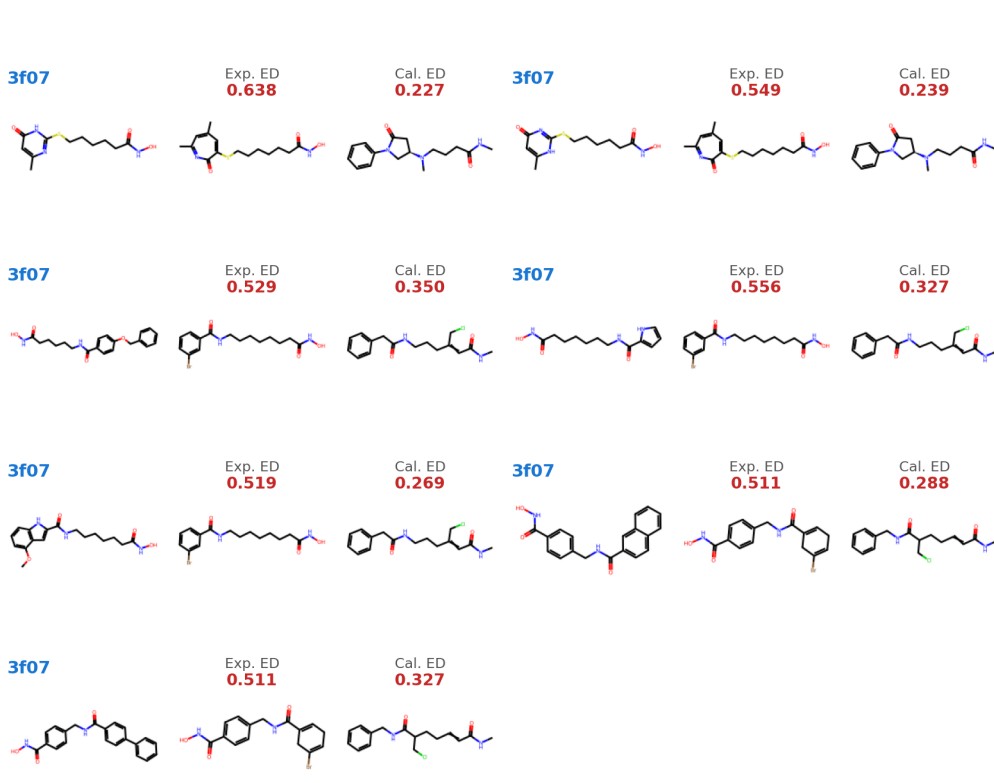

Figure 14: **Enhanced Fidelity of Molecular Recovery Using Experimental Electron Density (Part 4)**. This figure exemplifies a case where the structure refined against experimental electron density (middle) more accurately recapitulates the geometry of the active molecule (left) compared to its computationally-derived counterpart (right).

# C  PSEUDOCODE FOR ED-BFN GENERATION

Here, we present the generation procedure of ED-BFN. The complete process is summarized in the pseudocode 3.

---

**Algorithm 3** ED-BFN: High-Level Generation Pipeline

---

**Require:** Reference atoms $\mathcal{A}$, pharmacophore map $\phi$, density threshold $\rho_0$, steps $T$, mode $\in$ {oracle, soft}, optional $N_{\text{gt}}$, volume table $\mathcal{T}$

**Ensure:** Generated ligand $\mathcal{L}_T = \{(\mathbf{x}_k, \mathbf{h}_k)\}_{k=1}^N$

1: $\mathcal{P} \leftarrow$ Output of Algorithm 4
2: **if** mode == oracle **then**
3:     $N \leftarrow N_{\text{gt}}$
4: **else if** mode == soft **then**
5:     $V_{\mathcal{P}} \leftarrow$ ConvexHullVolume($\{\mathbf{p}_i\}$)
6:     Look up $(\mu_b, \sigma_b, a_b, b_b)$ from $\mathcal{T}$ using $V_{\mathcal{P}}$
7:     $N \sim \mathcal{N}_{[a_b,b_b]}(\mu_b, \sigma_b^2)$; clamp to $[1, N_{\max}]$
8: **end if**
9: Initialize $\boldsymbol{\theta}_1^{\mathcal{L}}$ for $N$ atoms (random positions, uniform types)
10: **for** $t = 1$ to $T$ **do**
11:     $\mathcal{G}_t \leftarrow$ Output of Algorithm 5
12:     $\{(\hat{\mathbf{x}}_k, \hat{\mathbf{h}}_k)\}_{k=1}^N \leftarrow \mathrm{F}_\theta(\mathcal{G}_t)$
13:     **for** $k = 1$ to $N$ **do**
14:         $y_k^{\mathbf{x}} \sim p_S(\cdot \mid \hat{\mathbf{x}}_k; \alpha_t)$, $y_k^{\mathbf{h}} \sim p_S(\cdot \mid \hat{\mathbf{h}}_k; \alpha_t)$
15:         $\boldsymbol{\theta}_{t+1,k}^{\text{type}} \leftarrow \mathrm{softmax}(\log \boldsymbol{\theta}_{t,k}^{\text{type}} + y_k^{\mathbf{h}})$
16:         $\boldsymbol{\mu}_{t+1,k} \leftarrow \boldsymbol{\mu}_{t,k} + \eta_t \cdot y_k^{\mathbf{x}}$
17:     **end for**
18: **end for**
19: **return** Sample: $\mathbf{x}_k \sim \mathcal{N}(\boldsymbol{\mu}_{T,k}, \lambda^{-1})$, $\mathbf{h}_k \sim \mathrm{Cat}(\boldsymbol{\theta}_{T,k}^{\text{type}})$

---

**Algorithm 4** Build Pharmacophore-Labeled Point Cloud $\mathcal{P}$

---

**Require:** Atomic coordinates $\mathcal{A} = \{\mathbf{a}_j\}$, pharmacophore map $\phi$, threshold $\rho_0$

**Ensure:** Point cloud $\mathcal{P} = \{(\mathbf{p}_i, l_i)\}_{i=1}^M$

1: Compute electron density $\rho(\mathbf{r})$ from $\mathcal{A}$ using cctbx
2: $\mathcal{P} \leftarrow \emptyset$
3: **for** $i = 1$ to $M$ **do**
4:     Sample $\mathbf{p}_i \sim \rho(\mathbf{r}) \cdot \mathbb{I}[\rho(\mathbf{r}) > \rho_0]$
5:     $\mathbf{a}^* \leftarrow \arg\min_{\mathbf{a}_j \in \mathcal{A}} \|\mathbf{p}_i - \mathbf{a}_j\|_2$
6:     $l_i \leftarrow \phi(\mathbf{a}^*)$
7:     Add $(\mathbf{p}_i, l_i)$ to $\mathcal{P}$
8: **end for**
9:
10: **return** $\mathcal{P}$

---

---

**Algorithm 5** Build Heterogeneous Interaction Graph $\mathcal{G}_t$

---

**Require:** Ligand params $\boldsymbol{\theta}_t^{\mathcal{L}} = \{\boldsymbol{\mu}_{t,k}, \dots\}_{k=1}^{N}$, point cloud $\mathcal{P} = \{(\mathbf{p}_i, l_i)\}_{i=1}^{M}$
**Ensure:** Graph $\mathcal{G}_t = (\mathcal{V}, \mathcal{E})$
1: Initialize node sets:
2: $\mathcal{V}_{\mathcal{P}} = \{v_i^{\mathcal{P}}\}$ with $\mathbf{f}_i^{\mathcal{P}} = \mathrm{MLP}_{\mathrm{emb}}(l_i)$
3: $\mathcal{V}_{\mathcal{L}} = \{v_k^{\mathcal{L}}\}$ with $\mathbf{f}_k^{\mathcal{L}} = \mathrm{concat}(\boldsymbol{\mu}_{t,k}, \dots)$
4: Initialize edge sets: $\mathcal{E}_{\mathcal{L}\leftrightarrow\mathcal{L}} \leftarrow \emptyset$, $\mathcal{E}_{\mathcal{P}\rightarrow\mathcal{L}} \leftarrow \emptyset$
5: **for** each ligand atom $v_k^{\mathcal{L}}$ **do**
6: $\mathcal{N}_k \leftarrow \mathrm{kNN}_{32}(\boldsymbol{\mu}_{t,k};$ all nodes in $\mathcal{V}_{\mathcal{P}} \cup \mathcal{V}_{\mathcal{L}})$
7: **for** each neighbor $v \in \mathcal{N}_k$ **do**
8:  **if** $v \in \mathcal{V}_{\mathcal{L}}$ **then**
9:   Add $(v_k^{\mathcal{L}}, v)$ to $\mathcal{E}_{\mathcal{L}\leftrightarrow\mathcal{L}}$
10:  **else if** $v \in \mathcal{V}_{\mathcal{P}}$ **then**
11:   Add $(v, v_k^{\mathcal{L}})$ to $\mathcal{E}_{\mathcal{P}\rightarrow\mathcal{L}}$ {Directed}
12:  **end if**
13: **end for**
14: **end for**
15:
16: **return** $\mathcal{G}_t = (\mathcal{V}_{\mathcal{P}} \cup \mathcal{V}_{\mathcal{L}}, \ \mathcal{E}_{\mathcal{L}\leftrightarrow\mathcal{L}} \cup \mathcal{E}_{\mathcal{P}\rightarrow\mathcal{L}})$

---

---

**Algorithm 6** ExpED: Ligand-Guided Experimental Density Sampling

---

**Require:** Reference ligand $\mathcal{L}_{\mathrm{ref}}$ (RDKit Mol), MTZ file $M$, CIF file $C$,
  density thresholds $\rho_{\min}, \rho_{\max}$ (in $\sigma$), max output $N_{\max}$

**Ensure:** Point cloud $\mathcal{P} = \{(\mathbf{p}_i, t_i)\}_{i=1}^N \subseteq \mathbb{R}^3 \times \{6, 7, 8, 9\}$

1: Parse unit cell $\mathcal{U}$ and space group $\mathcal{G}$ from $C$

2: Load $F$ and $\Phi$ from $M$; synthesize complex map coefficients if only amplitudes/phases given

3: Reassign Miller indices to crystal symmetry $(\mathcal{U}, \mathcal{G})$

4: Compute real-space electron density map $\mathbf{D}(\mathbf{r})$ via FFT with $\sigma$-scaling

5: Export $\mathbf{D}$ to XPLOR format; instantiate `XplorReader` $xr$

6: Extract heavy-atom positions $\{\mathbf{a}_j^{\mathrm{orig}}\}$ from $\mathcal{L}_{\mathrm{ref}}$

7: Wrap $\{\mathbf{a}_j^{\mathrm{orig}}\} \to \{\mathbf{a}_j^{\mathrm{wrap}}\}$ into $\mathcal{U}$: $[0, a] \times [0, b] \times [0, c]$

8: Update conformer of $\mathcal{L}_{\mathrm{ref}}$ to use $\{\mathbf{a}_j^{\mathrm{wrap}}\}$

9: Compute H/Donor/Acceptor (HDA) features:   $\mathcal{H} = \{(j, \mathbf{a}_j^{\mathrm{wrap}}, r_j^{\mathrm{vdW}}, \mathcal{F}_j)\}$  where $\mathcal{F}_j \in$
  $\{, \{\mathrm{Donor}\}, \{\mathrm{Acceptor}\}, \{\mathrm{Donor, Acceptor}\}\}$

10: Precompute ligand VdW envelopes: $\mathcal{V} = \{(\mathbf{a}_j^{\mathrm{wrap}}, 1.5 \cdot r_j^{\mathrm{vdW}})\}$

11: Initialize $\mathcal{P}_{\mathrm{raw}} \leftarrow$

12: **for** each grid voxel index $(i, j, k)$ in $\mathbf{D}$ **do**

13:  $\rho \leftarrow \mathbf{D}[i, j, k]$

14:  **if** $\rho_{\min} \leq \rho \leq \rho_{\max}$ **then**

15:   $\mathbf{p} \leftarrow xr.\texttt{idx2pos}([i, j, k])$ {fractional $\to$ Cartesian}

16:   **if** $\exists (\mathbf{a}_j^{\mathrm{wrap}}, r_j) \in \mathcal{V}$ s.t. $\|\mathbf{p} - \mathbf{a}_j^{\mathrm{wrap}}\| \leq r_j$ **and** no atom clash ($< 1.0\,\text{Å}$) **then**

17:    $\mathcal{F}_{\mathrm{near}} \leftarrow \{\mathcal{F}_j \mid \|\mathbf{p} - \mathbf{a}_j^{\mathrm{wrap}}\| < r_j^{\mathrm{vdW}} \wedge j \in \mathcal{H}\}$

18:    $t \leftarrow \begin{cases} 9 & \text{if } \exists j : \{\mathrm{Donor, Acceptor}\} \in \mathcal{F}_{\mathrm{near}} \\ 7 & \text{if } \mathrm{Donor} \in \bigcup \mathcal{F}_{\mathrm{near}} \wedge \mathrm{Acceptor} \notin \bigcup \mathcal{F}_{\mathrm{near}} \\ 8 & \text{if } \mathrm{Acceptor} \in \bigcup \mathcal{F}_{\mathrm{near}} \wedge \mathrm{Donor} \notin \bigcup \mathcal{F}_{\mathrm{near}} \\ 6 & \text{otherwise} \end{cases}$

19:    Probabilistically select one $t$ from candidates if multiple types present

20:    $\mathcal{P}_{\mathrm{raw}} \leftarrow \mathcal{P}_{\mathrm{raw}} \cup \{(\mathbf{p}, t)\}$

21:  **end if**

22: **end for**

23: Compute per-atom integer-cell shift vectors:  $\Delta_j \leftarrow$ nearest lattice vector to $(\mathbf{a}_j^{\mathrm{orig}} - \mathbf{a}_j^{\mathrm{wrap}})$

24: Build kd-tree over $\{\mathbf{a}_j^{\mathrm{wrap}}\}$

25: **for** each $(\mathbf{p}, t) \in \mathcal{P}_{\mathrm{raw}}$ **do**

26:  Find $j^* = \arg\min_j \|\mathbf{p} - \mathbf{a}_j^{\mathrm{wrap}}\|$ via kd-tree

27:  $\mathbf{p}_{\mathrm{unwrapped}} \leftarrow \mathbf{p} + \Delta_{j^*}$

28:  Append $(\mathbf{p}_{\mathrm{unwrapped}}, t)$ to $\mathcal{P}$

29: **end for**

30: **if** $|\mathcal{P}| > N_{\max}$ **then**

31:  $\mathcal{P} \leftarrow \texttt{FPS\_downsample}(\mathcal{P}, N_{\max}, \mathrm{preserve\_types} = \{7, 8, 9\})$

32: **end if**

33: **return** $\mathcal{P}$

---

---

**Algorithm 7** Unit-cell Wrapping of Ligand Coordinates

---

**Require:** Heavy-atom Cartesian coordinates $\{\mathbf{c}_j^{\text{orig}}\}_{j=1}^{M} \subseteq \mathbb{R}^3$,
   unit cell $\mathcal{U} = (a, b, c, \alpha, \beta, \gamma)$ (cctbx `unit_cell` object)
**Ensure:** Wrapped coordinates $\{\mathbf{c}_j^{\text{wrap}}\} \subseteq [0, a] \times [0, b] \times [0, c]$,
   per-atom integer-cell shift vectors $\{\boldsymbol{\Delta}_j\}$
1: Initialize $\{\mathbf{c}_j^{\text{wrap}}\} \leftarrow$, $\{\boldsymbol{\Delta}_j\} \leftarrow$
2: **for** $j = 1$ to $M$ **do**
3:    $\mathbf{f}_j \leftarrow \mathcal{U}.\texttt{fractionalize}(\mathbf{c}_j^{\text{orig}})$ {real $\rightarrow$ fractional}
4:    $\mathbf{f}_j^{\text{wrap}} \leftarrow (f_{j,x} \bmod 1,\ f_{j,y} \bmod 1,\ f_{j,z} \bmod 1)$ {into $[0,1)^3$}
5:    $\mathbf{c}_j^{\text{wrap}} \leftarrow \mathcal{U}.\texttt{orthogonalize}(\mathbf{f}_j^{\text{wrap}})$ {fractional $\rightarrow$ real}
6:    Append $\mathbf{c}_j^{\text{wrap}}$ to $\{\mathbf{c}_j^{\text{wrap}}\}$
7:    $\boldsymbol{\delta}_j \leftarrow \mathbf{c}_j^{\text{orig}} - \mathbf{c}_j^{\text{wrap}}$ {raw shift}
8:    $\mathbf{g}_j \leftarrow \mathcal{U}.\texttt{fractionalize}(\boldsymbol{\delta}_j)$
9:    $\mathbf{g}_j^{\text{int}} \leftarrow \texttt{round}(\mathbf{g}_j)$ {nearest lattice vector in frac. coords}
10:   $\boldsymbol{\Delta}_j \leftarrow \mathcal{U}.\texttt{orthogonalize}(\mathbf{g}_j^{\text{int}})$ {$\rightarrow$ real shift}
11:   Append $\boldsymbol{\Delta}_j$ to $\{\boldsymbol{\Delta}_j\}$
12: **end for**
13: **return** $\{\mathbf{c}_j^{\text{wrap}}\}$, $\{\boldsymbol{\Delta}_j\}$

---

**Note:** This procedure ensures compatibility between the ligand (often placed in the asymmetric unit, possibly with negative coordinates) and the P1-symmetrized electron density map (defined on the full unit cell). The shift vector $\boldsymbol{\Delta}_j$ is used later to "unwrap" sampled density points back into the original molecular frame.

---

---

**Algorithm 8** Uniform Downsampling via Farthest Point Sampling (FPS)

---

**Require:** Point cloud $\mathcal{P} = \{\mathbf{p}_i\}_{i=1}^N \subseteq \mathbb{R}^3$,
   optional types $\mathbf{t} = \{t_i \in \mathbb{N}\}_{i=1}^N$,
   target size $K < N$, flag $\texttt{preserve\_special} \in \{\texttt{True}, \texttt{False}\}$, seed $s \in \mathbb{Z}$
**Ensure:** Subset $\mathcal{S} = \{(\mathbf{p}_{i_k}, t_{i_k})\}_{k=1}^K, |\mathcal{S}| = K$
 1: $\mathcal{P} \leftarrow \texttt{array}(\mathcal{P})$; ensure $N > K$
 2: **if** $\texttt{preserve\_special}$ and $\mathbf{t}$ provided **then**
 3:    $\mathcal{I}_{\text{spec}} \leftarrow \{i \mid t_i \neq 6\}$ {e.g. donor/acceptor points}
 4:    $\mathcal{I}_{\text{gen}} \leftarrow \{i \mid t_i = 6\}$
 5:    $K_{\text{spec}} \leftarrow \min(|\mathcal{I}_{\text{spec}}|, K)$
 6:    $K_{\text{gen}} \leftarrow K - K_{\text{spec}}$
 7:    **if** $K_{\text{gen}} > 0$ **then**
 8:       $\mathcal{I}_{\text{selected}} \leftarrow \mathcal{I}_{\text{spec}} \cup \texttt{FPS\_subset}(\mathcal{P}[\mathcal{I}_{\text{gen}}], K_{\text{gen}}, s)$
 9:    **else**
10:       $\mathcal{I}_{\text{selected}} \leftarrow \mathcal{I}_{\text{spec}}$ (first $K$ items)
11:    **end if**
12: **else**
13:    $\mathcal{I}_{\text{selected}} \leftarrow \texttt{FPS\_subset}(\mathcal{P}, K, s)$
14: **end if**
15: $\mathcal{S} \leftarrow \{(\mathbf{p}_i, t_i) \mid i \in \mathcal{I}_{\text{selected}}\}$
16: **return** $\mathcal{S}$

**Subroutine** $\texttt{FPS\_subset}(\mathbf{X}, K, s)$:
1. Set random seed $s$; pick $i_1 \sim \text{Uniform}(\{1, \dots, N\})$
2. Initialize $D_j \leftarrow \infty$ for all $j$; set $D_{i_1} \leftarrow 0$
3. For $k = 2$ to $K$:
   • Update distances: $D_j \leftarrow \min(D_j, \|\mathbf{x}_j - \mathbf{x}_{i_{k-1}}\|)$
   • Choose $i_k \leftarrow \arg\max_j D_j$; set $D_{i_k} \leftarrow 0$
4. Return $\{i_1, \dots, i_K\}$

**Note:** FPS maximizes minimal inter-point distance, yielding spatially uniform coverage. When $\texttt{preserve\_special} = \texttt{True}$, all non-6 points (donor/acceptor, type 7/8/9) are retained first; remaining slots are filled by FPS on generic (type 6) points. This prevents loss of chemically critical features during downsampling.

---

# D FROM SCHRÖDINGER TO PHARMACOPHORE: WHY WE CHOOSE ELECTRON DENSITY AS OUR REPRESENTATIONS

We present a unified physical and mathematical framework demonstrating that **all representations commonly used in current drug design models, including electrostatic potential (ESP), pharmacophore features, and atomic coordinates, are lossy projections derived from the electron density field** $\rho_P(\mathbf{r})$. While this hierarchy may be self-evident to readers with a background in quantum chemistry, we explicitly formalize it as follows:

1. The **Schrödinger equation** yields $\rho_P$ as the fundamental quantum-mechanical observable, from which all molecular properties ultimately derive.

2. The **Poisson equation** computes the electrostatic potential (ESP) as a classical mean-field approximation of $\rho_P$, effectively integrating out quantum details.

3. Pharmacophore features emerge naturally from the **Hessian, gradient, and integrated occupancy of** $\rho_P$, without requiring atom typing or rule-based heuristics, offering a continuous, physics-grounded alternative to discrete pharmacophore definitions.

Together, this establishes $\rho_P$ as the **canonical, information-complete representation** for drug design — a foundational signal from which all other structural and energetic descriptors can, in principle, be derived.

## D.1 THE SCHRÖDINGER EQUATION GENERATES $\rho_P$ — THE ROOT REPRESENTATION

The electronic structure of a molecule is governed by the time-independent Schrödinger equation:

$$\hat{H}\Psi = \left( -\sum_{i=1}^{N} \frac{\nabla_i^2}{2} + \sum_{i<j} \frac{1}{|\mathbf{r}_i - \mathbf{r}_j|} + \sum_{i,A} \frac{Z_A}{|\mathbf{r}_i - \mathbf{R}_A|} \right) \Psi = E\Psi, \tag{1}$$

where $\Psi(\mathbf{r}_1, \ldots, \mathbf{r}_N)$ is the many-body wavefunction. In practice, for systems of drug-like molecules, direct solution is intractable. Density functional theory (DFT) circumvents this via the Hohenberg-Kohn theorems, replacing $\Psi$ with the electron density $\rho_P(\mathbf{r})$, and approximating many-body effects through an exchange-correlation functional $\hat{V}_{\text{xc}}[\rho]$ in the Kohn-Sham framework.

In practice, $\rho_P$ is computed via the Kohn-Sham equations, which express the density as a sum of single-particle orbital densities: $\rho_P(\mathbf{r}) = \sum_{i=1}^{N} |\phi_i(\mathbf{r})|^2$, rendering it computationally tractable for molecular systems.

The **electron density** is defined as the expectation of the density operator:

$$\rho_P(\mathbf{r}) = \langle \Psi | \hat{n}(\mathbf{r}) | \Psi \rangle = N \int |\Psi(\mathbf{r}, \mathbf{r}_2, \ldots, \mathbf{r}_N)|^2 d\mathbf{r}_2 \cdots d\mathbf{r}_N. \tag{2}$$

By the **Hohenberg-Kohn theorems**, $\rho_P(\mathbf{r})$ uniquely determines (up to a constant) the external potential — and thus all ground-state properties, including binding affinity, forces, and response functions. Therefore:

$\rho_P$ **is the minimal, complete, physics-grounded representation of molecular structure for drug design.**

## D.2 ESP IS A CLASSICAL, LOSSY PROJECTION OF $\rho_P$

The electrostatic potential $V_P(\mathbf{r})$ is not independent — it is derived from $\rho_P$ and nuclear charges via the **Poisson equation**:

$$\nabla^2 V_P(\mathbf{r}) = -4\pi \left( \rho_P(\mathbf{r}) - \sum_A Z_A \delta(\mathbf{r} - \mathbf{R_A}) \right), \tag{3}$$

whose solution is:

$$V_P(\boldsymbol{r}) = \int \frac{\rho_P(\boldsymbol{r}')}{|\boldsymbol{r} - \boldsymbol{r}'|} d\boldsymbol{r}' - \sum_A \frac{Z_A}{|\boldsymbol{r} - \mathbf{R_A}|}. \tag{4}$$

This constitutes a **linear, smoothing operator**: convolution with $1/r$ acts as a low-pass filter, attenuating high-frequency electronic features. Crucially, the ESP formulation ignores quantum many-body effects encoded in $\hat{V}_{\mathrm{xc}}[\rho]$, such as exchange correlation, Pauli repulsion, charge transfer, and quantum phase information. Consequently, it cannot distinguish between bonding and antibonding orbitals that yield identical $\rho_P$, nor capture directional anisotropies such as lone pairs or $\pi$-orbital orientations.

Thus, ESP is a **classical, mean-field approximation** of the true quantum electrostatic environment — useful, but incomplete.

### D.3    PHARMACOPHORE FEATURES AS EMERGENT PROPERTIES OF $\rho_P$: A RULE-FREE, PHYSICS-GROUNDED FORMALISM

We now demonstrate that **pharmacophore features can be not descirbed by heuristic rules but be geometric and electronic features naturally extractable from $\rho_P$ via its Hessian matrix and gradient field**.

### D.3.1    HESSIAN MATRIX ENCODES LOCAL ELECTRONIC CURVATURE

The Hessian of $\rho_P$ at point $\boldsymbol{r}$ is:

$$\mathbf{H}_\rho(\boldsymbol{r}) = \nabla^2 \rho_P(\boldsymbol{r}) = \begin{bmatrix} \partial_{xx}\rho & \partial_{xy}\rho & \partial_{xz}\rho \\ \partial_{yx}\rho & \partial_{yy}\rho & \partial_{yz}\rho \\ \partial_{zx}\rho & \partial_{zy}\rho & \partial_{zz}\rho \end{bmatrix}.$$

Its eigenvalues $\lambda_1 \geq \lambda_2 \geq \lambda_3$ and eigenvectors $\mathbf{e}_1, \mathbf{e}_2, \mathbf{e}_3$ describe the **local curvature** of the electron cloud:

1. **Positive eigenvalues** $\to$ convex bulges $\to$ regions of high electron occupancy (e.g., lone pairs on O, N);
2. **Negative eigenvalues** $\to$ concave depressions $\to$ regions of electron depletion (e.g., $\sigma$-holes on halogens, H-bond donor hydrogens);
3. **Anisotropic eigenvectors** $\to$ directional preference (e.g., lone pair orientation, $\pi$-orbital axis).

### D.3.2    GRADIENT FIELD $\nabla\rho_P$ ENCODES DIRECTIONALITY AND FLUX

The gradient $\nabla\rho_P(\boldsymbol{r})$ points in the direction of **steepest electron density increase**. In chemical contexts:

1. Near H-bond acceptors: $\nabla\rho_P$ points toward the lone pair lobe;
2. Near H-bond donors: $\nabla\rho_P$ points away from the hydrogen (toward electron-deficient region);
3. Near aromatic rings: $\nabla\rho_P$ oscillates perpendicular to the ring plane, encoding $\pi$-cloud anisotropy.

### D.3.3    PHARMACOPHORE FEATURE DETECTION AS A JOINT HESSIAN-GRADIENT FUNCTIONAL

We define a pharmacophore feature of type $l$ at location $\boldsymbol{p}^*$ as the solution to:

$$\boldsymbol{p}^* = \arg\max_{\boldsymbol{p} \in \Omega} \|\mathbf{H}_\rho(\boldsymbol{p}) \cdot \nabla\rho_P(\boldsymbol{p})\| \quad \text{subject to} \quad \nabla\rho_P(\boldsymbol{p}) \cdot \mathbf{n}_l > \tau_l, \tag{5}$$

where $\mathbf{H}_\rho \cdot \nabla \rho_P$ measures **curvature-aligned electronic flux** — peaks at lone pairs, $\sigma$-holes, and $\pi$-cloud edges; $\mathbf{n}_l$ is a fragment-frame normal enforcing chemical directionality (learnable or pre-defined); $\tau_l$ is a density-gradient threshold (e.g., high for Acceptor, low for Hydrophobic).

The feature type $l$ is then assigned based on the **signature of the Hessian eigenvalues**:

1. **Acceptor (e.g., carbonyl O)**: $\lambda_1 > 0, \lambda_2 \approx 0, \lambda_3 < 0 \rightarrow$ anisotropic bulge;
2. **Donor (e.g., OH/NH)**: $\lambda_1 < 0, \lambda_2 < 0, \lambda_3 \ll 0 \rightarrow$ concave depression;
3. **Hydrophobic (e.g., alkyl CH)**: All $\lambda_i \approx 0 \rightarrow$ flat, isotropic density;
4. **Aromatic**: Oscillating $\lambda_i$ signs + planar $\mathbf{e}_i \rightarrow \pi$-cloud signature.

### D.3.4 Tolerance $\epsilon$ as Conformational Entropy of Density Fluctuations

The pharmacophore tolerance $\epsilon$ is not a fixed radius — it is the **conformational entropy** associated with preserving the Hessian-gradient feature under thermal fluctuation:

$$\epsilon \propto -k_B \ln \int_{\mathcal{C}} \exp \left( -\beta \left\| \mathbf{H}_\rho(\boldsymbol{p}; \mathbf{c}) - \mathbf{H}_\rho^{\mathrm{ref}} \right\|_F^2 \right) d\mathbf{c}, \tag{6}$$

where $\mathcal{C}$ is the conformational ensemble, and $\| \cdot \|_F$ is Frobenius norm. This naturally explains why flexible sidechains have larger $\epsilon$ — their density curvature fluctuates more.

### D.4 A Unified Representation Learning Framework: All Methods Approximate Learning from $\rho_P$

In this appendix, we formalize the claim made in Section D: that all common structure-based representations are informationally subordinate to the electron density field $\rho_P(\mathbf{r})$. We begin by framing drug design as a representation learning problem.

Let $y \in \mathbb{R}$ denote a scalar bioactivity target (e.g., binding affinity). We posit that structure-based drug design (SBDD) can be fundamentally viewed as learning a representation $z = \phi(\cdot)$ such that the mapping $y \approx g_\theta(z)$ is maximally predictive. By the data processing inequality, predictive performance is upper-bounded by the mutual information between the representation and the target: $\mathcal{I}(z; y) \leq \mathcal{I}(\rho_P; y)$. Thus, an ideal representation should preserve as much of the physically relevant information in $\rho_P$ as possible.

As established in Section D, common SBDD representations — including electrostatic potential (ESP), pharmacophore features, and atomic coordinates — are all derived from $\rho_P$ via lossy physical projections. Empirical results (Table 1) further reveal that models relying on these approximations — while achieving high QED–SAS scores — often exhibit low docking scores and poor recovery of known active compounds. This suggests that such heuristic or geometrically simplified representations may be misaligned with the true bioactivity signal.

Crucially, $\rho_P$ is, by first principles, equivalent to the molecular conformational distribution under the Born–Oppenheimer approximation. It encodes not only nuclear positions but also electron delocalization, polarization, and non-covalent interaction potentials — making it the most natural starting point for learning bioactivity-relevant representations.

**Remark: On the "Directness" of Atomic Coordinates.** One might object that atomic-coordinate-based pocket representations — widely used in SBDD — are "more direct." However, we note that nearly all experimentally resolved protein structures (except those from NMR) are inferred from electron density maps (X-ray) or electrostatic potential fields (Cryo-EM). Moreover, these structures assume static, single-conformation models — discarding thermal motion and ensemble effects inherently captured in $\rho_P$. Thus, coordinate-based representations contain no more — and often less — information than their underlying density fields.

**Properties of Our Encoder $\phi_{\mathcal{P}}$.** It should be noted that our point-cloud encoder $\phi_{\mathcal{P}}$ is also lossy: we sample sparse points from $\rho_P$ and annotate them with pharmacophore labels, omitting explicit

intensity or gradient values. Nevertheless, as shown in Section D.3, pharmacophore features themselves are topological abstractions of critical points in $\rho_P$, derived from its Hessian and gradient. Therefore, our representation:

1. Strictly subsumes traditional pharmacophore models in expressivity (by retaining spatial point locations and pharmacophore semantics);

2. Inherits geometric and electronic structure directly from $\rho_P$, surpassing ESP- or atom-based encoders in physical fidelity;

3. Avoids the discretization and static assumptions inherent in grid- or coordinate-based methods.

**Summary.** While our labeled point cloud is not information-complete, it is — under the assumptions above — information-theoretically superior to existing alternatives. Observed performance gains (Table 1) can thus be attributed not merely to architectural choices, but to the richer physical grounding of the input representation.

## THE USE OF LARGE LANGUAGE MODELS

Large language models (LLMs) were used in this work solely as a post-hoc writing assistance tool to improve the clarity, grammar, and fluency of the manuscript's prose. All research ideas, experimental design, implementation, analysis, and intellectual contributions originated entirely from the authors. The LLM did not participate in any stage of ideation, hypothesis generation, data interpretation, or scientific reasoning.

The final content of this paper accurately reflects the authors' own understanding, background, and intentions. Authors take full responsibility for all claims, statements, and potential issues (including any inadvertent plagiarism or factual inaccuracies) in the submitted manuscript, regardless of LLM involvement in language polishing.

