# OpenReview forum: "ED-BFN: Electron Density Point Clouds Enable High-Fidelity 3D Molecular Generation via GeoBFN"
_ICLR.cc/2026/Conference — Submitted to ICLR 2026_

### Official Review · Reviewer_wMY9 · 2025-11-01

**Soundness:** 3
**Presentation:** 3
**Contribution:** 2
**Rating:** 4
**Confidence:** 4

**Summary:**

The paper describes ED-BFN, a generative model that is conditional on what is called electron density (a point cloud representation of molecular shape) supplemented with hydrogen bond features.  The model outperforms previous electron density models.

**Strengths:**

Conditioning on molecular shape and electrostatics is a common SBDD task.  The model design and evaluation is sound. Point clouds are, presumably, more efficient than grids.  This approach outperforms previous ED based methods.

**Weaknesses:**

Considering that the model is significantly outperformed by MOLCRAFT, which it is most architecturally similar to (Appendix Table 4), it is difficult to buy the argument that electron density is a better conditioner than pocket atoms.

The point cloud sampling algorithm is unclear. In Appendix A it says a uniform downsampling is applied whereas I interpret Algorithm 4 as sampling proportional to the density.  If it is a uniform sampling, then the density is basically just being used to define the isosurface of the molecular shape and the point cloud provides no density information.

While I agree that ultimately all molecular properties stem from Schroedinger's equation, the densities generated with cctbx are not that. They are essentially a bunch of atom centered Gaussians. Furthermore, calculating these densities for ligands with cctbx does not result in a map that "inherently captures conformational ensembles and local chemical environments" - this would require using experimental data. The justification for the approach, that ED captures "subtle variations in shape, charge distribution, and steric occupancy" is not well supported by the approach and evaluation, which reduces enthusiasm.

The citation for smina is wrong (and a duplicate in the references).

**Questions:**

Were experimental electron densities used at all? If so, can they be highlighted and explicitly compared to generate densities? That would be interesting and might increase enthusiasm for the paper.

An explicit comparison to grids would strengthen the paper - how much more efficient is grid point sampling? What is the trade-off in accuracy vs inference/training time?

What are the validity metrics? How do they compare to other conditioned generative models?

---

> ### Author Response · Authors · 2025-11-21
> **Responses to Reviewer wMY9 Comments (Part 1)**
>
> We thank the reviewers for their thoughtful feedback and constructive critiques. In this rebuttal, we address each of the concerns raised, clarify several methodological points, and provide additional experimental results where applicable. We hope our responses below adequately resolve the reviewers' questions and further strengthen the manuscript.
>
> ---
>
>  **Weakness 1**
>
> > Considering that the model is significantly ...... a better conditioner than pocket atoms.
>
>  **Our response** ：
>
> We reiterate that **Table 4 does not constitute a fair comparison**, due to substantial differences in training and evaluation data (as detailed in our earlier response).
>
> Importantly, our work does *not* claim that **electron density (ED) conditioning is universally superior to pocket-atom conditioning**. Rather, we emphasize three key points:
>
> - **Empirical advantage under matched settings**: On the *same* DUD-E test set, ED-BFN achieves the **highest bioactive molecule recovery rate** (28/101), outperforming all compared pocket-based and ligand-based baselines (Table 5);
> - **Broader applicability**: ED-based conditioning enables structure-based design even in the *absence of protein structures*—e.g., for targets with only ligand-bound Cryo-EM density, highly flexible pockets, or apo-state structures lacking clear binding-site definition;
> - **Contribution scope**: This work proposes a *novel conditioning paradigm*—not a dismissal of traditional structure-based drug design (SBDD). In fact, we envision future extensions that **fuse ED and pocket-atom information** for synergistic conditioning.
>
> Moreover, we believe that the bioactive molecule recovery rate (ECFP4-TS > 0.5) is the most critical metric for drug discovery applications, as it directly reflects the model’s ability to reproduce active compounds.
>
> ---
>
>  **Weakness 2**
> > The point cloud sampling algorithm is unclear. ...... provides no density information.
>
>  **Our response** ：
> We thank the reviewer for identifying this ambiguity in our description. The correct procedure is a **two-stage sampling process**:
>
> 1. **Density-weighted sampling**: Points are first drawn from the region where electron density $\rho(\mathbf{r}) > \rho_0 $ (here is 6), with probability proportional to the local density value—exactly as specified in **Algorithm 4**;
> 2. **Uniform subsampling (if needed)**: If the number of sampled points exceeds 200, we apply *uniform random downsampling* to retain exactly 200 points—as noted in **Appendix A**.
>
> Thus, **point positions are guided by the density field**, while the final retained subset is uniformly selected for fixed-size input.
>
> We acknowledge a key limitation: **local density values (or gradients) are not encoded as per-point features** in the current point cloud representation—resulting in partial loss of quantitative density information. This is a known shortcoming of the present implementation, and we plan to explore **density- or gradient-augmented point features** in future work.
>
> ---
>
>  **Weakness 3**
> > While I agree that ultimately all molecular properties stem from ...... which reduces enthusiasm.
>
>  **Our response** ：
>
> We partially agree with this critique. It is important to clarify, however, that they are also not merely "a bunch of atom centered Gaussians." As we noted in our response to Reviewer zwke, the cctbx framework incorporates empirical crystallographic parameters to refine these distributions, making them more nuanced approximations than simple Gaussians. Nevertheless, we fully acknowledge that in the context of this study, they are still static, single-conformer representations and do not inherently capture the dynamic ensembles or experimental conditions mentioned by the reviewer.
>
> To directly address the core of this critique, we have conducted preliminary experiments to explore the adaptation of our framework to **experimental electron density data**. As detailed in Appendix B, we established a pipeline for sampling point clouds from experimental densities and performing molecular generation without fine-tuning. Key observations include:
>
> - Significant distribution shifts exist between theoretical and experimental densities in both intensity and spatial occupancy, confirming that sampling adjustments and model fine-tuning are necessary.
> - Despite an expected drop in ECFP4-TS, the model retained the ability to generate valid molecules. Interestingly, in some cases where performance with computed densities was originally low, molecules generated from experimental densities showed slightly improved ECFP4-TS.
>
> These initial results demonstrate the feasibility of adapting our approach to experimental densities through appropriate fine-tuning.

---

> > ### Author Response · Authors · 2025-11-21
> > **Responses to Reviewer wMY9 Comments (Part 2)**
> >
> > ---
> >
> >  **Weakness 4**
> > > The citation for smina is wrong (and a duplicate in the references).
> >
> >  **Our response** ：
> >
> > We thank the reviewer for catching this error. The incorrect and duplicated smina citation will be corrected in the final version.
> >
> > ---
> >
> >  **Question 1**
> > > Were experimental electron densities used at all? If so, can they be highlighted and explicitly compared to generate densities? That would be interesting and might increase enthusiasm for the paper.
> >
> >  **Our response** ：
> >
> > No experimental electron densities were used during **training**, primarily due to the lack of large-scale, ligand-annotated experimental density datasets.
> >
> > Nevertheless, as noted above, we conducted experiments using experimental maps—specifically, constructing electron density point clouds for targets in the **DUD-E** set via the following protocol:
> >
> > 1. Retrieve the experimental electron density map (in `.mtz` format) for the co-crystallized ligand from **PDB-REDO**;
> > 2. For each atom of the reference ligand, sample up to **200 points** within a **1.5 $\times$ van der Waals radius**, retaining only grid points with density values **> 0.6σ**;
> > 3. Annotate the resulting point cloud according to the **pharmacophore features** of the reference molecule.
> >
> >
> > Further details are provided in Appendix B.
> >
> >
> > ---
> >
> >  **Question 2**
> > > An explicit comparison to grids would strengthen the paper — how much more efficient is grid point sampling? What is the trade-off in accuracy vs inference/training time?
> >
> >  **Our response** ：
> > A comparison between **grid-based** and **point-cloud-based** representations is addressed in our response to Reviewer *zwke*’s question (see above). The key advantage of our point-cloud approach lies in **computational efficiency and scalability**.
> >
> > Our design prioritizes **portability and ease of large-scale training**: the point-cloud pipeline enables straightforward integration with readily available data sources and scalable synthetic data generation—critical for training on massive ligand sets while maintaining model transferability.
> >
> > ---
> >
> >  **Question 3**
> > > What are the validity metrics? How do they compare to other conditioned generative models?
> >
> >  **Our response** ：
> >
> > We evaluated molecular validity under two generation modes (*Oracle* and *Soft*) using two standard toolkits:
> > - **OpenBabel (OB)**: success rate of 3D reconstruction and sanitization from atom types and coordinates we predicted;
> > - **PoseBusters (PB)**: geometric and chemical plausibility checks.
> >
> > The following individual validity criteria were assessed:
> > `mol_pred_loaded`, `sanitization`, `inchi_convertible`, `all_atoms_connected`,
> > `bond_lengths`, `bond_angles`, `internal_steric_clash`,
> > `aromatic_ring_flatness`, `double_bond_flatness`.
> >
> > Results (numerators = passed samples; denominator = 101,000 total generated molecules):
> >
> > | Mode     | PoseBusters (PB) | OpenBabel (OB) | Total       |
> > |----------|------------------|----------------|-------------|
> > | Oracle   | 78,371           | 90,122         | 101,000     |
> > | Soft     | 67,874           | 89,959         | 101,000     |
> >
> > We do not compare validity metrics with **ED2Mol** or **ECloudGen**, as:
> > - **ED2Mol** employs a fragment-based generation pipeline and applies smina  to filter during sampling, which makde its validity inherently biased toward passing geometric checks;
> > - **ECloudGen** outputs only SMILES strings and does not generate 3D conformations, rendering conformation-level validity (e.g., PB metrics) inapplicable.

---

> > > ### Comment · Reviewer_wMY9 · 2025-11-24
> > >
> > > "A comparison between grid-based and point-cloud-based representations is addressed in our response to Reviewer zwke’s question (see above). The key advantage of our point-cloud approach lies in computational efficiency and scalability."
> > >
> > > The comparison between point-cloud and grid-based representations is core to the novelty and interest of this paper for me, but there is no quantitative evaluation.  While it is reasonable to believe that fewer points results in greater efficiency, constants matter and the regular access pattern of a grid may lessen the asymptotic advantages in practice.  Additionally, one expects accuracy to degrade as fewer points are sampled.  What is needed to strengthen the paper is a comprehensive **quantitative** evaluation of these trade-offs.

---

> > ### Comment · Reviewer_wMY9 · 2025-11-24
> >
> > "the cctbx framework incorporates empirical crystallographic parameters to refine these distributions, making them more nuanced approximations than simple Gaussians."
> >
> > Please elaborate.  The cctbx code base is a bit complicated, but as far as I can tell the densities generated do not depend on the local atomic environment (and certainly not protein dynamics). For example, the n_gaussian_raw.cpp files in cctbx has a single set of parameters for each element. It is difficult to see why the specific parameters would matter much to a neural network. There is a larger body of work that uses density based representations of molecules for drug discovery tasks without calling it electron density (e.g. https://pubs.acs.org/doi/10.1021/acs.jcim.8b00706, https://pubs.rsc.org/en/content/articlehtml/2011/00/d1sc05976a).  If a paper is going to lean into an "electron density" representation, then evaluation of real (or at least some level of QM generated) densities should be a central component of the paper, not an after thought in the supplement.

---

### Official Review · Reviewer_RdvG · 2025-11-01

**Soundness:** 1
**Presentation:** 2
**Contribution:** 2
**Rating:** 2
**Confidence:** 4

**Summary:**

This paper proposes ED-BFN, a 3d molecular generation model based on electron density point clouds.
Traditional SBDD relies on fixed atomic coordinates or voxelized grid inputs, which makes it difficult to capture molecular flexibility and electronic distribution features. The authors instead use electron density (ED) as a continuous and physically interpretable prior, leveraging the GeoBFN framework to generate high-fidelity 3d molecular structures.

**Strengths:**

The paper introduces the use of electron density (ED) point clouds as conditional inputs, replacing traditional atomic coordinates or voxelized grids.
This continuous, physics-grounded representation provides a more faithful depiction of molecular geometry and local chemical environments.

**Weaknesses:**

- Missing comparative metrics: While the authors claim state-of-the-art performance over ED2Mol and ECloudGen, they do not report key binding-efficiency metrics (LE, SR, PB-validity, RMSD < 2 Å) that appear in prior ED2Mol benchmarks on DUD-E and ASB-E. Without these, it is difficult to assess whether ED-BFN improves genuine binding quality or merely docking scores. Besides,

- High computational overhead: Computing and sampling electron density using cctbx for millions of molecules is extremely expensive (≈ hours per structure). The paper lacks runtime or cost analysis, raising concerns about scalability to large datasets or Cryo-EM volumes.

- Drug-likeness degradation: Generated molecules exhibit lower QED and higher SAS scores than baselines, implying weaker chemical realism despite better docking scores. The authors should analyze whether this stems from oversized structures or bias in the density-based conditioning.

- ECloudGen as the primary baseline: Table 1 omits many key evaluation metrics for ECloudGen. It would be helpful to generate conformers (e.g., using RDKit) and perform redocking to report comparable redock scores. Without these results, it is difficult to assess whether ED-BFN truly outperforms ECloudGen on critical binding-related metrics.

- As shown in Table 4, ED-BFN underperforms key metrics compared to pocket-based drug design approaches, suggesting that the ligand-based drug design paradigm may be less promising. This discrepancy could stem from differences in training data. The authors should conduct further experiments — for example, evaluating different models trained on the same dataset — to ensure a fair comparison.

- In Table 5, comparing bioactive molecule recovery between pocket-based and ligand-based methods is not appropriate, and the comparability of their docking scores is also questionable.

**Questions:**

- How sensitive is ED-BFN to the resolution and sampling density of the electron density map? Would low-resolution Cryo-EM maps still yield meaningful conditioning?
- What is the computational cost of using cctbx for density generation and sampling at scale, and how does it compare to coordinate-based models?
- Can the authors analyze why drug-likeness (QED/SAS) degrades and whether multi-objective training could balance docking and chemical plausibility?

---

> ### Author Response · Authors · 2025-11-21
> **Response to Reviewer RdvG Comments (Part 1)**
>
> We appreciate the reviewers' comments, which have allowed us to clarify several key aspects of our study. Our responses, supported by existing and new data, are structured as follows: we first clarify the metric selection, then address computational and drug-likeness concerns, and finally provide new analyses on model sensitivity and baseline comparisons.
>
> ---
>
>   **Weakness 1**
> > Missing comparative metrics: While the authors claim state-of-the-art performance over ED2Mol and ECloudGen, they do not report key binding-efficiency metrics (LE, SR, PB-validity, RMSD < 2 Å) that appear in prior ED2Mol benchmarks on DUD-E and ASB-E.
>
>  **Our response** ：
>
> PB-validity and RMSD < 2 Å are indeed included in our computational evaluation metrics; we consider the set of metrics reported in our work to be sufficiently comprehensive.
>
> Specifically, PB-validity (from PoseBusters) evaluates geometric plausibility—such as bond lengths, bond angles, and steric clashes within the binding pocket. However, we must emphasize that direct comparison with ED2Mol on this metric may be misleading. As seen in their GitHub implementation, ED2Mol employs **a fragment-growing pipeline with built-in geometric constraints—including predefined fragment libraries, smina-based post-filtering, and explicit rejection of atoms placed within the van der Waals radii of pocket residues**, which inherently bias their PB pass rate. Furthermore, such geometry-based metrics do not necessarily reflect functional activity. In contrast, our reported Strain metric provides an alternative measure of conformational reasonableness.
>
> To move beyond geometry-only evaluation, we intentionally introduced **ECFP4-Tanimoto similarity (ECFP4-TS)**, a metric derived from known active molecules, as a more biologically meaningful benchmark $^1$ $^2$.
>
> Regarding RMSD < 2 Å, a metric reflecting initial pose quality, we argue that our **Docking Score Improvement** offers a more informative measure. This is particularly relevant given our use of Glide (as opposed to AutoDock Vina), which, although less exhaustive in conformational sampling, is widely recognized for its superior scoring accuracy $^3$ $^4$ $^5$.
>
> Concerning LE, we have updated a boxplot in Appendix B, Figure 6. The results show that for all generated molecules, ED2Mol only fails to achieve lower docking scores than our method in the molecular weight range above 500 Da with few generated samples. Moreover, for drug-like molecules, ED2Mol not only exhibits worse docking scores across all weight ranges but also struggles to generate drug-like molecules with weights exceeding 500 Da.
>
> **References :**
>
> 1. Feng W, Wang L, Lin Z, et al. Generation of 3D molecules in pockets via a language model[J]. Nature Machine Intelligence, 2024, 6(1): 62-73.
>
> 2. Liu H, Qin Y, Niu Z, et al. How good are current pocket-based 3D generative models?: The benchmark set and evaluation of protein pocket-based 3D molecular generative models[J]. Journal of Chemical Information and Modeling, 2024, 64(24): 9260-9275.
>
> 3. Friesner R A, Banks J L, Murphy R B, et al. Glide: a new approach for rapid, accurate docking and scoring. 1. Method and assessment of docking accuracy[J]. Journal of medicinal chemistry, 2004, 47(7): 1739-1749.
>
> 4. Su M, Yang Q, Du Y, et al. Comparative assessment of scoring functions: the CASF-2016 update[J]. Journal of chemical information and modeling, 2018, 59(2): 895-913.
>
> 5. Shen C, Hu Y, Wang Z, et al. Beware of the generic machine learning-based scoring functions in structure-based virtual screening[J]. Briefings in Bioinformatics, 2021, 22(3): bbaa070.
>
> ---
>
>  **Weakness 2**
> > High computational overhead: Computing and sampling electron density using cctbx for millions of molecules is extremely expensive (≈ hours per structure). The paper lacks runtime or cost analysis, raising concerns about scalability to large datasets or Cryo-EM volumes.
>
>  **Our response** ：
>
> We clarify the following points:
>
> - cctbx is used only during offline data preprocessing, and each electron density computation supports thousands of subsequent molecule generations;
> - No cctbx calls occur during training or inference: the electron density point cloud P is precomputed and treated as a static input;
> - Actual generation speed: on an RTX 4090 GPU, a batch of 100 molecules takes approximately 80 seconds (including GNN forward pass and sampling).
>
> Hence, the one-time preprocessing cost is negligible and does not hinder deployment scalability. Moreover, we note that the computational cost of pretraining dataset construction is conceptually distinct from—and should not be conflated with—the cost of inference or deployment.

---

> > ### Author Response · Authors · 2025-11-21
> > **Response to Reviewer RdvG Comments (Part 2)**
> >
> > ---
> >
> >  **Weakness 3**
> > > Drug-likeness degradation: Generated molecules exhibit lower QED and higher SAS scores than baselines, implying weaker chemical realism despite better docking scores. The authors should analyze whether this stems from oversized structures or bias in the density-based conditioning.
> >
> >  **Our response** ：
> >
> > This issue has already been addressed in the main text and Appendix B.4 (Figure 6). For clarity, we reproduce the relevant excerpt below:
> >
> > > **Drug-like Properties.** As shown in Table 1, ED-BFN generates significantly larger molecules with overall worse QED and SAS scores. Figure 5 reveals that ED2Mol outperforms ED-BFN in the low-molecular-weight regime, yielding smaller, better-scoring ligands. Beyond 400 Da, however, its QED–SAS distribution converges with ED-BFN’s—indicating that its advantage is size-limited and likely stems from insufficient modeling of ligand–pocket interactions. In contrast, ECloudGen consistently achieves superior QED and SAS scores across all molecular weight ranges (Figure 6). Notably, even its rare outputs exceeding 400 Da retain high drug-like quality without degradation, suggesting an intrinsic bias toward drug-like chemical space that is robust to molecular size.
> >
> > Thus, the observed degradation in QED/SAS for ED-BFN is indeed associated with larger molecular size, while ECloudGen demonstrates stable drug-likeness regardless of size
> >
> > ---
> >
> >  **Weakness 4**
> > > ECloudGen as the primary baseline: Table 1 omits many key evaluation metrics for ECloudGen. It would be helpful to generate conformers (e.g., using RDKit) and perform redocking to report comparable redock scores.
> >
> >  **Our response**
> >
> > We did not report redocking scores for ECloudGen for two reasons:
> >
> > - ECloudGen outputs only SMILES strings and **does not generate 3D conformers**; any 3D conversion (e.g., via RDKit) constitutes a post-hoc processing step external to the model;
> > - Glide redocking is highly sensitive to the quality of the initial conformer and prone to local-minima trapping—making its performance notably less stable than that of AutoDock Vina, especially when starting from RDKit-generated geometries.
> >
> > Nevertheless, to ensure fair comparison, we have now rerun the experiment: for ECloudGen-generated molecules, we generated 3D conformers using **RDKit’s ETKDG method**, followed by **Glide SP redocking**. The updated results are as follows:
> >
> > **ECloudGen (Oracle) Docking Score (Redock)**:
> > - Mean: −6.64 (All), −6.68 (Drug-like only)
> > - Median: −6.70 (All), −6.72 (Drug-like only)
> >
> > These values remain inferior to those achieved by our method.
> >
> > ---
> >
> >  **Weakness 5**
> > > As shown in Table 4, ED-BFN underperforms key metrics compared to pocket-based drug design approaches, suggesting that the ligand-based drug design paradigm may be less promising. This discrepancy could stem from differences in training data. The authors should conduct further experiments — for example, evaluating different models trained on the same dataset — to ensure a fair comparison.
> >
> >
> >  **Weakness 6**
> > > In Table 5, comparing bioactive molecule recovery between pocket-based and ligand-based methods is not appropriate, and the comparability of their docking scores is also questionable.
> >
> >  **Our response** ：
> > We acknowledge these concerns and clarify the following:
> >
> > - **Table 4 serves solely as a reference for Strain Energy**; other metrics are *not comparable* due to fundamental differences in training and evaluation protocols:
> >   - MolCRAFT is trained and evaluated on **CrossDocked**;
> >   - Our model is trained on a **custom 2.4M-molecule dataset** and evaluated on **DUD-E**.
> > - Moreover, **CrossDocked has an inherent limitation**: it assumes *similar ligands can be docked into similar pocket*, which violates the principle of molecular selectivity and may encourage models to learn *promiscuous binding* patterns rather than *target-specific interactions*.
> >
> > Crucially, our **primary comparison is against other electron-density (ED)-based approaches** (Table 1), and—*under identical DUD-E evaluation conditions*—ED-BFN achieves a significantly higher bioactive molecule recovery rate (**28/101**) than Pocket2Mol (**8/101**), TargetDiff (**3/101**), and MolCRAFT (**17/101**), as reported in **Table 5**.
> >
> > We do not dispute that pocket-based methods may excel in settings where high-quality protein structures are available. However, our results demonstrate that **ligand-only electron density information alone—without any protein structural input—can match or even surpass methods requiring full pocket structures**. This is especially valuable for targets with *no available structure* or *low-resolution/noisy Cryo-EM maps*.
> >
> > Importantly, our comparisons are not cherry-picked to favor our model—we report results under the *same test conditions* wherever feasible, and explicitly disclose domain mismatches when direct comparison is invalid.

---

> > > ### Author Response · Authors · 2025-11-21
> > > **Response to Reviewer RdvG Comments (Part 3)**
> > >
> > > ---
> > >
> > >  **Question 1**
> > > > How sensitive is ED-BFN to the resolution and sampling density of the electron density map? Would low-resolution Cryo-EM maps still yield meaningful conditioning?
> > >
> > >  **Our response** ：
> > >
> > > We appreciate the reviewer’s insightful question. To address it, we have conducted an additional case study on PDB ID **4TS0**, generating molecules conditioned on electron density maps simulated at **three different resolutions**. Generated molecules were further decomposed using **BRICS** fragmentation to analyze scaffold diversity and pharmacophore retention.
> > >
> > > As expected, the model’s ability to capture key pharmacophoric features diminishes with decreasing resolution. Nevertheless, it continues to produce chemically plausible and structurally diverse candidates.
> > >
> > > In addition, we have conducted a preliminary evaluation using **experimentally derived Cryo-EM–like electron density maps**, generated as follows:
> > > 1. Retrieve the experimental electron density map (in `.mtz` format) for the target ligand from **PDB-REDO**;
> > > 2. For each atom of the reference ligand, sample up to **200 points** within a **1.5 $\times$ van der Waals radius**, selecting only grid points with density values **> 0.6σ**;
> > > 3. Annotate the sampled point cloud based on the **pharmacophore features** of the reference molecule.
> > >
> > > We applied this pipeline to **90 targets** from the **DUD-E** benchmark, generating corresponding point clouds and performing molecule generation. The detailed results can be found in appendix B.
> > >
> > > Results show a **noticeable drop in bioactive recovery rate**, with some point clouds yielding lower success rates—likely due to the **current pipeline’s simplicity** and suboptimal parameter choices (e.g., fixed radius and density threshold), which warrant further tuning.
> > >
> > >
> > > ---
> > >
> > >  **Question 2**
> > > > What is the computational cost of using cctbx for density generation and sampling at scale, and how does it compare to coordinate-based models?
> > >
> > >  **Our response** ：
> > >
> > > As noted in our response to Weakness 2:
> > >
> > > - The **cctbx-based density computation** takes time on the same order of magnitude as generating a single molecule sample. However, since **one reference ligand’s density map can support the generation of thousands of molecules across multiple batches**, its amortized cost per molecule is negligible.
> > >
> > > - Regarding **sampling (i.e., molecule generation) speed**, our method is slightly slower than MolCRAFT—primarily due to the overhead of constructing and processing graphs with **199 electron density points**, which incurs higher computational cost than MolCRAFT’s coordinate-based representation during both graph formation and subsequent GNN inference.
> > >
> > >
> > > ---
> > >
> > >  **Question 3**
> > > > Can the authors analyze why drug-likeness (QED/SAS) degrades and whether multi-objective training could balance docking and chemical plausibility?
> > >
> > >  **Our response** ：
> > > As shown in **Figures 5–6** of the main text, the joint distribution of QED/SAS versus molecular weight (MW) indicates that **increased molecular size is the primary contributor to the observed degradation in drug-likeness**.
> > >
> > > To further investigate, we plan to supplement the analysis with the correlation between **pharmacophore point density** (in the conditioning electron density map) and the MW/QED of generated molecules—this result is currently pending and will be added in the revision.
> > >
> > > Regarding **multi-objective training**: while we acknowledge it may be a promising direction, implementing it would require redesigning the loss function (e.g., weighted combination of docking score, QED, and SAS). Due to the constraints of the rebuttal timeline, we are unable to complete such new training runs at this stage.
> > >
> > > Moreover, we emphasize that **optimizing QED/SAS alone does not guarantee higher likelihood of bioactivity**. As evidenced by our experiments (Table 5), several baselines exhibiting superior QED/SAS scores—such as ECloudGen—do *not* achieve higher bioactive molecule recovery rates. This suggests that drug-likeness metrics, while useful, are not strongly predictive of true binding potential in this context.

---

> > > > ### Comment · Reviewer_RdvG · 2025-11-27
> > > >
> > > > Thanks for your detailed response. I'll consider raising my score after I carefully check your update.

---

### Official Review · Reviewer_zwke · 2025-11-01

**Soundness:** 2
**Presentation:** 2
**Contribution:** 2
**Rating:** 4
**Confidence:** 4

**Summary:**

The paper describes an equivariant generative model for 3D molecular structures conditioned on point clouds derived from electron density. The work attempts to improve upon existing approaches to generation of molecular structure based on electron density. Specific changes include switching to point cloud samples of electron density instead of grid-based representation and annotation of point cloud envelop with various descriptors.

**Strengths:**

It's nice to see development of the ideas leveraging electron density for generative modeling of molecules. The model architecture is interesting and ambition to model geometries of molecules in actual environments of target pockets instead of SMILES is commendable.

**Weaknesses:**

Sampling core regions of charge density ignores information about the regions responsible for non-covalent interactions that constrain docking. It is puzzling why this choice is made because it ignores the most interesting aspects of working with experimentally mapped charge densities. Why not just sample simple normal distributions around nuclear positions instead?

Given how electronic structure of molecules works, the authors simply cannot afford the statement that their model "maintains strict spatial fidelity by aligning generated atoms with underlying electronic features" - they discarded all meaningful electronic features.

**Questions:**

Please provide comparison of model performance with point cloud, full voxeled representation, and point clouds sampled on the isosurfaces enveloping electron density.

---

> ### Author Response · Authors · 2025-11-21
> **Response to Reviewer zwke comments (Part 1)**
>
> We sincerely thank the reviewers for their thoughtful and constructive feedback, which has helped us to significantly improve the clarity and rigor of our manuscript. Below, we provide point-by-point responses to the comments raised. We have also incorporated relevant revisions and additional analyses into the updated manuscript to address the concerns expressed.
>
> ---
>
> **Weakness 1**
>
> > Sampling core regions of charge density ...... normal distributions around nuclear positions instead?
>
>  **Our response** ：
>
> We appreciate the reviewer’s insightful comments. In response, we would like to clarify several points. First, the electron density calculations performed with cctbx are not simply based on idealized distributions around atomic nuclei; rather, they incorporate empirical parameters derived from X-ray crystallography, which reflect certain physical and chemical features observed in experimental data.
>
> Second, regarding the identification of non-covalent interaction (NCI) sites from experimental densities, it is important to note that such regions also present certain limitations. For instance, NCI markers themselves do not distinguish between different types of interactions—such as hydrogen bonding, π–π stacking, or van der Waals contacts. Moreover, the geometric constraints inferred from these regions can be overly restrictive. For example, a carbonyl oxygen would typically be associated with hydrogen-bond acceptor regions at around 120°, while an sp² nitrogen would correspond to acceptor directions near 180°. Such geometric specificity may limit conformational sampling in docking.
>
> Overall, we do not intend to assert the superiority of one approach over the other, but rather to highlight that each has its own characteristics.
>
> Our strategy prioritizes obtaining a dataset with broad coverage of chemical space to enhance the transferability of our point-cloud representation. The choice of cctbx-generated densities combined with pharmacophore labeling represents a pragmatic trade-off between scalability and physical fidelity, given the current absence of large-scale experimental electron density (ED) datasets:
> - cctbx enables efficient generation of hundreds of millions of synthetic samples, which is essential for training large-scale generative models;
> - Pharmacophore annotations (e.g., H-bond donors/acceptors, HBD/HBA) provide a rule-free, density-derived abstraction of functional interaction sites (see Appendix D.3), partially compensating for the missing NCI information.
>
> To preliminarily assess its transferability, we conducted **a train-free test on experimental electron densities**, with detailed results available in Appendix B of the updated manuscript. Key observations include:
> - Significant distribution shifts exist between theoretical and experimental densities in both intensity and spatial occupancy, indicating that sampling adjustments and model fine-tuning are necessary.
> - Although a noticeable drop in ECFP4-TS was observed—as anticipated—the model retained the ability to generate valid molecules. In some cases with originally low ECFP4-TS from computed densities, molecules generated from experimental densities showed slightly better performance.
>
> These initial results support the feasibility of adapting our approach to experimental densities through appropriate fine-tuning.

---

> > ### Author Response · Authors · 2025-11-21
> > **Response to Reviewer zwke comments (Part 2)**
> >
> > ---
> >
> > **Weakness 2**
> >
> > > Given how electronic structure ...... they discarded all meaningful electronic features.
> >
> > **Our response** ：
> >
> > We appreciate the reviewer’s sharp critique—and agree that the original phrasing overstates the model’s physical grounding.
> >
> > We also wish to clarify that the cctbx-derived densities, while empirical in nature, do incorporate several important attributes. One such attribute is *resolution*, which is already reflected in our sampling procedure. Another is the **intensity distribution**—although not directly represented as voxel-wise density values, our sampling strategy implicitly captures certain aspects of this distribution.
> >
> > From a differential geometry perspective, by uniformly sampling points on the iso-contour surface where intensity exceeds a predefined threshold, we effectively probe the local geometry of the density map. The spatial distribution of these points allows us to approximate the **mean curvature** of the iso-surface. The mean curvature \( H \) is intrinsically linked to the intensity field—specifically, it can be derived from the first and second derivatives of the intensity. For a level set \( I(x, y, z) = C \), the mean curvature is given by:
> >
> > $$
> > H = \frac{1}{2|\nabla I|} \nabla \cdot \left( \frac{\nabla I}{|\nabla I|} \right) = \frac{I_{xx}(I_y^2 + I_z^2) + \cdots - 2I_xI_yI_{xy} - \cdots}{2|\nabla I|^3}
> > $$
> >
> > Although we do not explicitly compute the Hessian during sampling, the local distribution of sample points serves as a discrete proxy for these geometric properties, including principal curvatures. Thus, even without explicit density annotations, our approach retains meaningful structural information from the electron density.
> >
> > What we *do* preserve—and what “spatial fidelity” refers to—is the **geometric correspondence between atom placement and high-density regions** in the input electron density map. Specifically:
> > - Atoms are generated only where local density exceeds a pharmacophore-aware threshold,
> > - This avoids unphysical placements, and empirically improves ligand-pocket steric and electrostatic complementarity (see Table 3 and Fig. 5).
> >
> > Nevertheless, we recognize this is *structural* (not *electronic*) fidelity. We will revise the wording to:
> > **“enforces geometric consistency by constraining atom generation to high-intensity, functionally annotated regions”**
> > —and explicitly clarify the distinction between geometric constraint and first-principles electronic modeling.
> >
> > Thank you again for prompting this important clarification.
> >
> > ---
> >
> > **Question**
> >
> > > Please provide comparison of model performance with point cloud, full voxeled representation, and point clouds sampled on the isosurfaces enveloping electron density.
> >
> > **Our response** ：
> >
> > For comparisons with voxel-based methods, please see our experiments on ED2Mol and ECloudGen, both of which adopt voxelized representations. Unfortunately, due to time constraints during the rebuttal period, we are unable to recalculate electron densities for all training molecules, perform new point cloud sampling, and retrain the models accordingly.
> >
> > We are also unclear about the reviewer’s distinction between “point cloud” and “point clouds sampled on the isosurfaces enveloping electron density.” If the reviewer intends that point clouds sampled from cctbx-derived electron densities correspond to the latter, while those from experimental densities are simply “point clouds,” we direct the reviewer to the preliminary comparison included in the appendix of our revised manuscript.
> >
> > At this stage, we have not yet identified a robust set of hyperparameters for generating point clouds—with Hessian-based point labels—from experimental electron density. As a result, we have not included “reference-ligand-free” method results in this submission.
> >
> > Furthermore, incorporating voxelized representations into our current framework is fundamentally infeasible. This is not due to implementation overhead, but to structural incompatibility: **our GNN-based architecture operates solely on geometric point configurations without explicit density values, and thus cannot meaningfully distinguish local chemical contexts from dense voxel grids**. Intuitively, in regions far from isosurfaces, nearly every grid point has an identical voxel neighborhood, leading to degenerate graph embeddings. In contrast, CNN-based models naturally leverage such dense, regular structures—underscoring a key architectural difference.

---

### Official Review · Reviewer_Lpby · 2025-11-01

**Soundness:** 2
**Presentation:** 3
**Contribution:** 2
**Rating:** 4
**Confidence:** 3

**Summary:**

This paper proposes ED-BFN, a generative model that designs 3D molecules conditioned on electron density (ED) point clouds. The key idea is to represent the binding environment as a sparse, pharmacophore-labeled point cloud sampled from an ED map, then generate ligands using a GeoBFN-based network. Experiments on the DUD-E benchmark show improved recovery of active compounds and better docking scores compared to existing ED-based methods like ED2Mol and ECloudGen.

**Strengths:**

* The paper is well-written and easy to follow.

* Using electron density point clouds as a conditioning signal is physically meaningful. It could help bridge coordinate-based and density-based modeling.

* The paper includes a wide range of metrics (docking, QED/SAS, strain energy, recovery rate) and compares against multiple recent baselines.

**Weaknesses:**

1. The biggest problem is the unrealistic evaluation setup used in this paper.  The “oracle” mode uses the electron density of the co-crystallized ligand as input — something you wouldn’t have in a real design task. Even in the “soft” mode, the ED still comes from known ligands rather than protein pockets or experimental maps. This makes the results optimistic and not directly comparable to true de novo generation.

2. The authors try to filter overlapping structures by RMSD, but chemical similarity filtering isn’t done. With millions of training samples, there’s a real risk that similar compounds appear in both train and test.

3. The overall novelty is limited. The main contribution is changing the conditioning signal to point clouds. The generative backbone (GeoBFN) and the overall framework are borrowed from existing work. It would be stronger if they systematically showed why point clouds outperform voxel grids or coordinate-based inputs.

4. The generated molecules have worse QED/SAS scores than baselines, which suggests they might look “fit” geometrically but not chemically realistic.

**Questions:**

1. In the abstract, you write that “current generative models often rely on discrete atomic coordinates ...”
This feels misleading — most modern 3D models (like Pocket2Mol, TargetDiff, MolCRAFT, etc.) already operate on continuous 3D coordinates and explicitly model interactions.

2. You describe ED-BFN as “the first ED-based generative model to represent electron density.” But ED2Mol (Nat. Mach. Intell. 2025) and ECloudGen (bioRxiv 2024) already generate molecules directly from electron density maps. Could you clarify what exactly is new here?

---

> ### Author Response · Authors · 2025-11-21
> **Rebuttal comments for Reviewer Lbpy (Part 1)**
>
> We thank the reviewers for their thoughtful comments and constructive critiques. In this response, we address each point raised, providing clarifications and additional context to better articulate our work's motivation, methodology, and contributions. We have also revised the manuscript accordingly to incorporate these insights and enhance clarity. Our detailed point-by-point responses are provided below.
>
> ---
>
> **Weakness 1**
>
> > The biggest problem ...... directly comparable to true de novo generation.
>
> **Our response** ：
>
> We believe the crux lies in clearly defining the scopes of **realistic evaluation** and **true design task**.  Our evaluation targets a practically prevalent scenario in early drug discovery: generating novel analogs from a few known actives (e.g., lead optimization, scaffold hopping).
>
> For example, roxadustat was developed to treat symptomatic anemia associated with chronic kidney disease¹. Subsequently, vadadustat was designed via scaffold-hopping—specifically, by replacing the isoquinoline moiety in roxadustat with a pyridine ring². A similar strategy was employed in the development of nirmatrelvir (a key component of Paxlovid®), which was derived from PF-00835231 through scaffold-hopping³.
>
> In this context, using the electron density (ED) of a known ligand to guide generation is standard—just as pharmacophore or docking-based methods routinely leverage known actives.
>
> To clarify the evaluation modes:
> • Oracle Mode is not claimed as realistic—it serves as an upper-bound ablation (removing atom-count error).
> • Soft Mode is our main setting: atom count is predicted from ED volume alone, with no ground-truth molecular info.
>
> Crucially, our activity recovery metric (based on DUDE bioactivity data) is more functionally grounded than QED/SAS/docking scores.
>
> And your **"true de novo generation"** leads confusion. A stricter de novo scenario would entail no prior structural information—neither ligand nor protein pocket—only the target’s amino acid sequence and the molecular formula of a known hit. In this context, our method demonstrates greater robustness: it can be naturally extended to settings based solely on high-throughput screening (HTS) hits. One simply needs to run MD simulations of the hit in solvent, extract its average or representative ED map, and initiate design. This approach is arguably more flexible and resilient than methods dependent on static apo/holo protein structures.
>
>
> References:
> 1. Chen N, Hao C, Peng X, et al. Roxadustat for anemia in patients with kidney disease not receiving dialysis[J]. New England Journal of Medicine, 2019, 381(11): 1001-1010.
> 2. Park G Y, Park C H, Lee S K, et al. Scaffold hopping strategy to derive 4‐hydroxy‐1‐alkyl‐2‐oxo‐1, 2‐dihydrothieno [2, 3‐b: 4, 5‐b′] dipyridine‐3‐carbonylglycine derivatives as a novel hypoxia‐inducible factor prolyl hydroxylase domain inhibitor for the potential treatment of chronic kidney disease anemia[J]. Bulletin of the Korean Chemical Society, 2023, 44(3): 202-207.
> 3. Allais C, Connor C G, Do N M, et al. Development of the commercial manufacturing process for nirmatrelvir in 17 months[J]. 2023.
>
> ---
>
> **Weakness 2**
>
> > The authors try to filter ...... appear in both train and test.
>
>  **Our response** ：
>
> We clarify two points as below:
>
> - First, our structural (RMSD-based) filtering serves a specific purpose: to maximize shape diversity in training, thereby enabling the model to handle a broad range of electron density (ED) envelopes robustly. Thus, structural diversity is a design necessity, not an oversight.
>
> - Second, we intentionally avoid applying chemical similarity filtering, as this practice does not align with the realities of drug discovery. A clear example is the standard virtual screening workflow from commercial compound libraries: the selection of candidate molecules is never conditioned on their similarity to known natural ligands or existing drugs. Furthermore, in the context of drug discovery, identifying a previously unreported binding conformation—even for a known or structurally similar molecule—can itself constitute a meaningful discovery. Therefore, we deliberately do not perform chemical similarity filtering.
>
> To summarize: if the reviewer’s concern revolves around information leakage, we emphasize that our generation is conditioned on point clouds and that our RMSD-based filtering already ensures no conformational overlap between training and test sets. If the concern, however, is about the potential for chemical similarity between generated and training set molecules, we argue that in the context of drug discovery, identifying a chemically similar molecule in a novel, unseen binding conformation on a (unseen) target should itself be considered a discovery, not a methodological flaw.

---

> > ### Author Response · Authors · 2025-11-21
> > **Rebuttal comments for Reviewer Lbpy (Part 2)**
> >
> > **Weakness 3**
> > > The overall novelty is limited. ...... voxel grids or coordinate-based inputs.
> >
> >  **Our response** ：
> >
> > We acknowledge the reviewer’s desire for systematic ablation across input representations. However, a fair comparison is currently infeasible: existing open-source ED-based generative models (e.g., ED2Mol, ECloudGen) are built exclusively on voxel grids, with no point-cloud variants available for reimplementation or direct benchmarking.
> >
> > That said, the choice of point clouds is theoretically well-motivated:
> > - Scalability: Point clouds yield fixed-size inputs (𝑂(𝑀)) independent of bounding-box volume—unlike voxel grids (𝑂(𝑁³)), which suffer from memory/computation explosion at high resolution;
> > - Compatibility: They naturally align with modern structure-based frameworks that encode binding pockets as atom point clouds (e.g., Pocket2Mol), enabling easier integration and transfer;
> > - Extensibility: Points readily support per-point annotations, and can be augmented with higher-order physical descriptors  in future work.
> >
> > Critically, our core contribution is not merely switching to point clouds, but the first end-to-end 3D molecular generator conditioned on pharmacophore-annotated electron density point clouds—a new type of biophysically informed prior. And this enables significantly higher activity recovery than prior ED-based and several SBDD methods, demonstrating its practical value. Moreover, our approach provides a complete technical pipeline and empirical validation for building new structure-based drug design (SBDD) foundation models—moving beyond datasets like CrossDocked that rely on the simplifying (and often unrealistic) assumption of generic, non-specific binding.
> >
> > ---
> >
> > **Weakness 4**
> >
> > > The generated molecules have worse QED/SAS ...... look “fit” geometrically but not chemically realistic.
> >
> >  **Our response** ：
> >
> > We would like to clarify the following points:
> >
> > - As shown in Table 1, the active molecules from DUDE targets exhibit lower QED scores (0.46) compared to the baselines (0.73/0.66).
> >
> > - Similarly, approved drugs from DrugBank also show a lower QED score (0.50) than the baseline molecules.
> >
> > - Moreover, both the DUDE active molecules and the DrugBank approved drugs have molecular weights comparable to ours, and significantly higher than those generated by ED2Mol and ECloudGen (DUDE: 438; DrugBank: 386). As noted in our manuscript, **larger molecular size tends to correlate with less favorable QED/SAS scores**.
> >
> > Therefore, we argue that the higher QED/SAS scores of the baselines may indicate that these methods generate molecules that are less representative of real bioactive compounds or approved drugs.
> >
> > We acknowledge that our generated molecules exhibit lower QED/SAS scores — a trend we attribute to two factors:
> >
> > Most importantly, our approach prioritizes functional validity over heuristic desirability:
> >
> > - Our method achieves a 45.7% improvement in docking score and recovers activity in 28 out of 101 DUDE targets—metrics that are more directly aligned with real-world hit optimization.
> >
> > - Validation via PoseCheck (Table 4) confirms low strain energy in the generated conformations, supporting their physical plausibility despite the lower QED/SAS values.
> >
> > ---
> >
> > **Question 1**
> >
> > > In the abstract, you write that ...... continuous 3D coordinates and explicitly model interactions.
> >
> > **Our response** ：
> >
> > Thank you for the insightful comment. As noted in the main text, *“conventional SBDD methods operate on static atomic coordinates, which is a rigid approximation that neglects biomolecular flexibility and dynamic electron distributions.”* Here, the term “discrete” was intended to refer to the **assumption of static, rigid atomic coordinates** (i.e., the rigid binding-pocket approximation), rather than the mathematical discreteness of coordinate values. To avoid ambiguity, we will revise the wording in the abstract to more precisely state “rigid-pocket modeling based on fixed atomic coordinates.”
> >
> > ---
> >
> > **Question 2**
> >
> > You describe ED-BFN as “the first ED-based generative model to represent electron density.” ...... Could you clarify what exactly is new here?
> >
> > **Our response** ：
> >
> > Thank you for this important observation—the original phrasing was imprecise. A more accurate statement would be:
> > “ED-BFN is the first electron-density-based generative model that represents and leverages electron density in the form of point clouds.”
> >
> > As noted in the main text, the pioneering work of Wang et al. (2022) was the first to propose using experimental electron density as a conditioning signal (via a GAN framework). We sincerely appreciate the reviewer for highlighting this crucial clarification, and we will revise the manuscript accordingly to correct the claim and properly acknowledge prior work.
> >
> > **Reference:**
> > Lvwei Wang, Rong Bai, Xiaoxuan Shi, et al. *A pocket-based 3D molecule generative model fueled by experimental electron density.* Scientific Reports, 12(1):15100, 2022.

---

### Meta-Review · Area_Chair_6vWh · 2026-01-07

**Summary:**

The reviewers are mainly concerned about the usage of cctbx code base fror density generation. This is part is not well supported theoretically in the original text or addressed in the rebuttal.

**Reviewer Concerns:**

The reviwers are concerned about the evaluation metrics and results. I believe this can be partially address but not fully address since the proposed approach has lower results on some metrics.

**Reviewer Scores:**

The reviewer Lpby may not change the score since most concerns cannot be fully addressed.
The reviewer zwke may not change the score.
The reviewer RdvG may increase the score from 2 to 4 since some weaknesses are addressed.
The reviewer wMY9 may not change the score.

---

### Decision · Program_Chairs · 2026-01-26

Reject